# Biochemical characterization of the cyclooxygenase enzyme in penaeid shrimp

**Punsa Tobwor[1], Pacharawan Deenarn[1], Thapanee Pruksatrakul[1], Surasak Jiemsup[1], Suganya Yongkiettrakul[1], Vanicha Vichai[1], Metavee Phromson[1], Sage Chaiyapechara[1], Waraporn Jangsutthivorawat[1], Pisut Yotbuntueng[1], Oliver George Hargreaves[2], Wananit Wimuttisuk[1]***

1 National Center for Genetic Engineering and Biotechnology (BIOTEC), National Science and Technology Development Agency (NSTDA), Khlong Luang, Pathum Thani, Thailand, 2 School of Biosciences, University of Kent, Canterbury, Kent, United Kingdom

* wananit.wim@biotec.or.th

**Data Availability Statement:** All relevant data are within the paper and its Supporting Information files.

## Abstract

Cyclooxygenase (COX) is a two-step enzyme that converts arachidonic acid into prostaglandin $H_2$, a labile intermediate used in the production of prostaglandin $E_2$ ($PGE_2$) and prostaglandin $F_{2\alpha}$ ($PGF_{2\alpha}$). In vertebrates and corals, COX must be *N*-glycosylated on at least two asparagine residues in the N-(X)-S/T motif to be catalytically active. Although COX glycosylation requirement is well-characterized in many species, whether crustacean COXs require *N*-glycosylation for their enzymatic function have not been investigated. In this study, a 1,842-base pair *cox* gene was obtained from ovarian cDNA of the black tiger shrimp *Penaeus monodon*. Sequence analysis revealed that essential catalytic residues and putative catalytic domains of *P. monodon* COX (PmCOX) were well-conserved in relation to other vertebrate and crustacean COXs. Expression of PmCOX in 293T cells increased levels of secreted $PGE_2$ and $PGF_{2\alpha}$ up to 60- and 77-fold, respectively, compared to control cells. Incubation of purified PmCOX with endoglycosidase H, which cleaves oligosaccharides from *N*-linked glycoproteins, reduced the molecular mass of PmCOX. Similarly, addition of tunicamycin, which inhibits *N*-linked glycosylation, in PmCOX-expressing cells resulted in PmCOX protein with lower molecular mass than those obtained from untreated cells, suggesting that PmCOX was *N*-glycosylated. Three potential glycosylation sites of PmCOX were identified at N79, N170 and N424. Mutational analysis revealed that although all three residues were glycosylated, only mutations at N170 and N424 completely abolished catalytic function. Inhibition of COX activity by ibuprofen treatment also decreased the levels of $PGE_2$ in shrimp haemolymph. This study not only establishes the presence of the COX enzyme in penaeid shrimp, but also reveals that *N*-glycosylation sites are highly conserved and required for COX function in crustaceans.

## Introduction

Prostaglandins serve as signaling molecules that regulate various physiological processes, including inflammation, immune response, blood clotting and reproduction [1–4]. Two

**Funding:** The study was supported by the following: W.W. Grant Number RSA6180007, Thailand Science Research and Innovation https://www.tsri.or.th/; P.T. Grant Number P1650758, Research Initiative Grant from the National Center for Genetic Engineering and Biotechnology; W.W. Grant Number P1750566, Research and Platform Technology Management Grant from the National Center for Genetic Engineering and Biotechnology. The funders had no role in study design, data collection and analysis, decision to publish, or preparation of the manuscript.

**Competing interests:** The authors have declared that no competing interests exist.

isoforms of mammalian COX enzymes, namely COX1 and COX2, are considered constitutive and inducible enzymes, respectively [5]. COX is responsible for regulating the entry of substrates into the prostaglandin biosynthesis pathway by catalyzing a two-step reaction that converts arachidonic acid (ARA) to prostaglandin $H_2$ ($PGH_2$) [5–7]. The $PGH_2$ then serves as the first intermediate for the production of downstream prostaglandins, including prostaglandin $E_2$ ($PGE_2$) and prostaglandin $F_{2\alpha}$ ($PGF_{2\alpha}$) [8–10].

Studies in mammals have revealed that COX is a 70-kDa enzyme with three conserved domains, namely an epidermal growth factor domain, a membrane-binding domain and a catalytic domain [7, 11–13]. The center of the catalytic domain contains heme, which is anchored by proximal and distal histidines inside the active site [14–16]. COX also contains at least four N-(X)-S/T N-glycosylation motifs. It has been shown that N-glycosylation mediates protein folding and enzymatic function of COX [7, 17–19]. In both mammalian COX1 and COX2, glycosylation at N68, N144 and N410 is essential for protein folding and catalytic function while glycosylation of COX2 at N580 regulates protein stability [17–19].

Although COX is highly conserved in vertebrates, the presence of COX is evolutionarily diverse in invertebrate species. Two COX isoforms, namely COX-A and COX-B, have been identified in the Arctic soft coral *Gersemia fruticosa*, while only one COX isoform has been identified in the Caribbean gorgonian *Plexaura homomalla* [20, 21]. In crustaceans, full-length *cox* genes have been identified in the fresh water flea *Daphnia pulex*, the black tiger shrimp *Penaeus monodon*, the Pacific white shrimp *Penaeus vannamei*, the American lobster *Homarus americanus*, and amphipod crustaceans *Caprella* sp. and *Gammarus* sp. [22–26]. The presence of a COX enzyme has been proposed in the giant tiger prawn *Macrobrachium rosenbergii* based on positive immunostaining of prawn ovaries with an anti-COX1 antibody [27]. Currently, COX enzymatic function has only been verified in the amphipod crustaceans *Gammarus* sp. and *Caprella* sp. [23]. Varvas *et al.* (2009) demonstrated that these amphipod COX enzymes were N-glycosylated as tunicamycin treatments reduced COX molecular masses in both species [23]. Nevertheless, the locations of these N-glycosylation sites and their involvement in COX catalytic activity have yet to be determined.

In this study, the catalytic function of *P. monodon* COX (PmCOX) was established by determining the levels of prostaglandins in cells expressing the PmCOX enzyme. The N-glycosylation requirement for PmCOX was also examined using endoglycosidase digestion, tunicamycin treatment and site-directed mutagenesis. Lastly, ibuprofen treatment was used to demonstrate catalytic activity of the COX enzymes in *P. monodon* and *P. vannamei* in *in vitro* and *in vivo* inhibition assays, respectively.

## Materials and methods

### Ethical statement

Shrimp trials and experiments in this study were approved by the Institutional Animal Care and Use Committee of the National Center for Genetic Engineering and Biotechnology, Thailand (BT-Animal 28/2560). All experiments were performed in accordance with Animal Research: Reporting of *In Vivo* Experiments (ARRIVE) and conformed with international and national legal and ethical requirements [28, 29].

### RNA extraction and cDNA synthesis

Shrimp ovary samples were subjected to total RNA extraction using Trizol® reagent (Thermo Fisher Scientific, Massachusetts, USA). The RNA concentration was estimated using the NanoDrop 2000 spectrophotometer (Thermo Fisher Scientific). Ovarian cDNA was

synthesized using the RevertAid[TM] First Strand cDNA Synthesis Kit with oligo (dT) 18 primer (Thermo Fisher Scientific) according to the manufacturer's instructions.

## Gene amplification and plasmid construction

The full-length *Pmcox* gene (GenBank: KF501342.1) was amplified from *P. monodon* ovarian cDNA using *Pmcox*-HindIII-F and *Pmcox*-XbaI-R primers. Primer sequences for *Pmcox* cloning are shown in Table 1. The full-length *Pmcox* gene was cloned into pcDNA3.1[TM] (+)/*myc*-His B (Thermo Fisher Scientific), resulting in a plasmid construct encoding the full-length PmCOX protein with a Myc-His tag at the C-terminus. Sequencing was performed (1[st]-BASE, Malaysia) to confirm the correct sequence in the plasmid [30].

## Sequence analysis

Protein BLAST (BLASTP version 2.10.0[+]) analysis was performed to verify the identity of the protein [30]. AU-rich elements (AREs) and the AAUAAA motif in the 3′ untranslated region (3′ UTR) were analyzed using the PolyAPred algorithm [31]. PmCOX conserved domains were predicted using the Conserved Domain Architecture Retrieval Tool (CDART) program [32]. Multiple sequence alignment between PmCOX and its homologs was performed using the Clustal Omega program [33]. A phylogenetic tree was constructed using the Molecular Evolutionary Genetics Analysis (MEGA) version 10.1 program (https://www.megasoftware.net/) using the maximum likelihood method [34]. An *N*-glycosylation motif search was performed using NetNGlyc 1.0 prediction (www.cbs.dtu.dk/services/NetNGlyc/) [35]. An *N*-glycosylation potential of 0.5 was considered as the minimum cut-off value.

## Site-directed mutagenesis

To remove potential glycosylation sites on PmCOX, site-directed mutagenesis was performed using KAPA HiFi PCR kit mutagenesis (Kapa Biosystems, Massachusetts, USA) for N-to-Q substitutions at amino acid residues 79, 170 and 424. Single, double and triple mutations at the glycosylation sites of PmCOX were generated. Primer sequences used for the PCR mutagenesis are shown in Table 1. The restriction enzyme *Dpn*I (Thermo Fisher Scientific) was used to remove the template plasmid prior to transformation into *E. coli* DH5α. Transformants were selected for plasmid extraction using colony PCR. Recombinant plasmids harboring mutated *Pmcox* were verified by DNA sequencing (1[st]-BASE) and Clustal Omega multiple sequence alignment against wild-type PmCOX [33].

**Table 1. Primer sequences for *Pmcox* cloning and site-directed mutagenesis.**

| Constructs | Primers | Primer sequences (5′–3′) | Tm (°C) |
|---|---|---|---|
| pcDNA3.1[TM](+)*Pmcox*/*myc*-His B | *Pmcox*-HindIII-F | GCAAGCTTATGTCAACGTCTGTGTTGAAAACCA | 68 |
| | *Pmcox*-XbaI-R | GCTCTAGACTAGGGGGTTCCTTGCGGGA | |
| N79Q mutant | *Pmcox*N79Q-F | CAAGCCCGACCGACAATACACATGCGACTG | 65 |
| | *Pmcox*N79Q-R | CAGTCGCATGTGTATTGTCGGTCGGGCTTG | |
| N170Q mutant | *Pmcox*N170Q-F | CTCAATGCCTACTACCAAGAGAGCTTCTACGGC | 60 |
| | *Pmcox*N170Q-R | GCCGTAGAAGCTCTCTTGGTAGTAGGCATTGAG | |
| N424Q mutant | *Pmcox*N424Q-F | ATTCCCGATACTCTCCAAGTGAGCGGGACAGAT | 65 |
| | *Pmcox*N424Q-R | ATCTGTCCCGCTCACTTGGAGAGTATCGGGAAT | |

Underlined letters indicate changes in nucleotide sequences to create an N-to-Q substitution.

## PmCOX expression in mammalian cells

Human embryonic kidney 293T cells (ATCC ® CRL-11268[TM]; ATCC, Virginia, USA) were maintained in Dulbecco's modified Eagle's medium (DMEM), supplemented with 10% (v/v) fetal bovine serum (FBS) and 1% (v/v) penicillin-streptomycin solution (Invitrogen, New York, USA). Cells were incubated at 37˚C in 5% $CO_2$ and subcultured every three days. For transient protein expression, expression plasmids containing wild-type PmCOX or PmCOX glycosylation mutants, the empty vector (pcDNA3.1[TM](+)/*myc*-His B), or pEGFP-N1 (negative control) were transiently transfected into 293T cells by calcium phosphate precipitation [36]. Culture media were replaced 18 h post-transfection and the transfected cells were harvested at 48 h to assess protein expression by Western blotting.

## Characterizing PmCOX enzymatic function

Cells were transfected with expression plasmids encoding wild-type PmCOX or PmCOX glycosylation mutants. At 48 h post-transfection, fresh DMEM without FBS was added to the transfected cells. Cells were incubated with either 0.1% dimethyl sulfoxide (DMSO; vehicle) or 10–40 µM ARA (Cayman Chemical, Michigan, USA) for 30 min at 37˚C. Cells were then harvested for Western blot analysis to verify protein expression. Cell culture media were collected to determine levels of secreted $PGE_2$ by enzyme immunoassay (EIA). The remaining media were subjected to C18 solid phase extraction (SPE) and analyzed using ultra-performance liquid chromatography high-resolution tandem mass spectrometry (UPLC-HRMS/MS). Five replicates were used for each experimental condition.

## Determining levels of $PGE_2$ using EIA

Levels of secreted $PGE_2$ were determined using the $PGE_2$ EIA kit (Cayman Chemical), according to the manufacturer's instructions. Dose-response curves were generated using $PGE_2$ standards ranging from 15.63–250.00 pg/mL. The limit of quantification for $PGE_2$ was 15.63 pg/mL. Intra- and inter-day coefficients of variance (CV) were 6.41% and 7.57%, respectively.

## C18 SPE

Culture media harvested from PmCOX-expressing cells were adjusted to pH 4.0 using formic acid. Ten percent (w/v) butylated hydroxytoluene (Sigma-Aldrich, Missouri, USA) in ethanol and prostaglandin $B_1$ ($PGB_1$) were added to the media as an antioxidant and an internal standard, respectively. Ten milliliters of cell culture media were loaded onto Vertipak[TM] C18 SPE cartridges (Vertical Chromatography, Bangkok, Thailand), which had previously been washed with 10 mL methanol and 10 mL water. Sample-loaded cartridges were washed twice with 5 mL water and eluted twice with 5 mL ethyl acetate. Eluates were evaporated and reconstituted in ethanol for the identification and quantification of prostaglandins by UPLC-HRMS/MS.

## UPLC-HRMS/MS analysis

**Standard mixtures.** The standard mixture used in this study consisted of 10 polyunsaturated fatty acids (PUFA) and eicosanoid compounds, comprising 70.5 µM $PGF_{2\alpha}$, 500 µM $PGE_2$, 500 µM $PGB_1$, 149.5 µM 15d-prostaglandin $J_2$ (15d-$PGJ_2$), 62.4 µM 8-hydroxyeicosapentaenoic acid, 62.8 µM 12(R)-hydroxyeicosatetraenoic acid (12(R)-HETE), 62.8 µM 5-HETE, 90 µM eicosapentaenoic acid (EPA), 75 µM docosahexaenoic acid (DHA) and 25 µM ARA. PUFA and eicosanoid standard compounds were individually dissolved in 100% ethanol and stored as stock solutions at –80˚C. The concentration of each standard solution was adjusted to achieve similar ionization intensity signals under UPLC-HRMS/MS. The

calibration curve of $PGF_{2\alpha}$ and $PGE_2$ was established with a linearity range of 1.95–125 nM and 15.6–1000 nM, respectively (S1 Table). Because other forms of eicosanoids, such as 15d-$PGJ_2$, HETEs and HEPEs, were not detected in cell culture medium by UPLC-HRMS/MS, the calibration curves for these standard compounds were not established. All extracts and standards were analyzed by UPLC-HRMS/MS under parallel reaction monitoring (PRM)-based targeted mass spectrometry with a negative ion mode.

**UPLC-HRMS/MS conditions.** UPLC-HRMS/MS was performed using the DIONEX 3000 RS UPLC system coupled to the Orbitrap Fusion™ Tribrid™ mass spectrometer. Conditions for liquid chromatography included auto-sampler temperature at 10°C, column temperature at 40°C, and injection volume at 5 μL. SPE extracts were separated using an Acclaim® 120 C18 column (2 μm, 2.1 mm x 150 mm) (Dionex, Surrey, UK). The mobile phase consisted of (A) 0.01% (v/v) acetic acid in water and (B) 0.01% (v/v) acetic acid in acetonitrile, with a flow rate of 300 μL/min for a total run time of 23 min. For the analysis of prostaglandins from cell culture media, the gradient program started with 30% solution B and increased to 100% solution B within 17 min at a flow rate of 0.3 mL/min. The mobile phase was held at 100% solution B for 1 min and returned to 30% solution B within 0.5 min. The mobile phase was maintained at 30% solution B for 4.5 min for column re-equilibration. On the other hand, the gradient used for shrimp tissue samples was adjusted for better resolution. Elution was conducted with a linear gradient starting from 30% B to 70% B within 17 min, then rising to 100% B within 1 min. Subsequently, the elution gradient was returned to the starting condition of 30% B within 0.5 min. Finally, 30% B was held for 4.5 min before the next injection.

Mass spectrometer conditions were set with electrospray ionization voltage at 2,500 V in negative mode. Nitrogen gas was used as the sheath gas at 40 psi and as the auxiliary gas at 12 psi. Helium was used as the collision gas with ion transfer tube temperature at 333°C. The vaporizer temperature was 317°C. Ion products of PUFA and eicosanoid standards were analyzed at a resolution of 120,000 with a 5e4 automatic gain control (AGC) target and the maximum injection time was set at 246 ms. A scheduled PRM was used for PUFA and eicosanoid analysis. Analytical characteristics of $PGF_{2\alpha}$ and $PGE_2$ are provided in S1 Table.

## Tunicamycin treatment

Five micrograms of pcDNA3.1™(+)*Pmcox*/*myc*-His B were transfected into 293T cells. Forty-eight hours after transfection, cells were washed with phosphate-buffered saline (PBS) and incubated with complete DMEM supplemented with 10% FBS spiked with 2 μg/ml of tunicamycin (Sigma-Aldrich, Missouri, USA) or 0.1% DMSO (vehicle). Cells were incubated at 37°C for 16 h in a $CO_2$ incubator before being harvested and stored at –20°C for Western blot analysis.

## Western blot analysis

Transfected 293T cells were harvested and sonicated for 10 s in radioimmunoprecipitation assay (RIPA) buffer (1% Nonidet™ P-40, 1% sodium deoxycholate, 1% sodium dodecyl sulfate, 150 mM sodium chloride, 0.01 M sodium phosphate buffer pH 7.2 and 2 mM EDTA) containing a cOmplete™ EDTA-free protease inhibitor cocktail (Sigma-Aldrich). Protein concentrations from cell lysates were determined using the Lowry assay [37]. Forty micrograms of protein samples were separated by electrophoresis on 10–12% polyacrylamide gels, transferred to Hybond® ECL nitrocellulose membrane (GE healthcare, Illinois, USA) and incubated with anti-His monoclonal IgG antibodies (Abcam, Massachusetts, USA) for 1 h. Proteins were visualized using horseradish peroxidase-conjugated secondary antibodies (Vector Laboratories,

California, USA) and SuperSignal West Pico Chemiluminescent Substrate$^{TM}$ (Thermo Fisher Scientific).

## Immunoprecipitation assay

Three hundred micrograms of cell lysate containing PmCOX-Myc-His was incubated with 2 μL of anti-His antibodies (Abcam) with gentle rocking at 4˚C for 1 h. Sixty microliters of protein A Sepharose beads CL-B4 (GE Healthcare, Illinois, USA) was added to RIPA buffer to prepare a 50% slurry before addition to the cell lysate mixture. The mixture was incubated with gentle rocking at 4˚C for 1 h. Protein-bound Sepharose beads were spun down and washed twice in 1 mL RIPA buffer.

## Endoglycosidase H digestion of PmCOX

Immunoprecipitated PmCOX was digested with endoglycosidase H enzyme (Promega, Wisconsin, USA) according to the manufacturer's instruction. Briefly, protein A Sepharose beads bound to PmCOX proteins were incubated in denaturing solution at 95˚C for 5 min. Endoglycosidase H enzyme was added to the denatured protein and the mixture was incubated at 37˚C for 18 h.

## *In vitro* inhibition assay using *P. monodon* haemolymph

*P. monodon* haemolymph was drawn from the ventral sinus at the first pair of pleopods from 10 juvenile *P. monodon* with an average body weight of 12 g. The haemolymph was mixed with 10% ice-cold sodium citrate (anti-coagulant) at a 1:1 ratio (v/v) of haemolymph:anti-coagulant. The haemolymph was pooled, divided into 1-mL aliquots, and incubated with 15 or 150 ng/mL aspirin, 20 or 200 ng/mL indomethacin, and 20 or 200 ng/mL ibuprofen, based on effective dosages published in other crustaceans [38–40]. After a 30-min incubation, 0.1% DMSO or 10 μM ARA was added to the haemolymph and the mixture was incubated at 28˚C, with shaking at 200 rpm for 30 min. Haemolymph was then spun down at 1,500 ×g for 5 min at 4˚C to remove haemocytes and other cell debris. PGE$_2$ levels in the haemolymph were determined using the EIA kit. The experiment was performed in triplicate to determine percent inhibition of each COX inhibitor. Levels of PGE$_2$ in haemolymph samples were subtracted by the basal levels of PGE$_2$ obtained from the samples treated with 0.1% DMSO (negative control), with 100% PGE$_2$ production defined as levels of PGE$_2$ from haemolymph treated with 10 μM ARA (positive control). Percentages of PmCOX inhibition under inhibitor treatment were determined as follows:

$$\% \text{ inhibition} = \frac{\text{PGE2 level}_{\text{positive control}} - \text{PGE2}_{\text{inhibitor}}}{\text{PGE2 level}_{\text{positive control}}} \; x \; 100$$

## *In vivo* inhibition assay in *P. vannamei*

Juvenile *P. vannamei* with a body weight range of 10–12 g (*n* = 12) were obtained from commercial culture farms in Prachuap Khiri Khan province, Thailand. The shrimp were tested to be free from specific pathogens (Taura syndrome virus, white spot syndrome virus, yellow head virus, and infectious hypodermal and hematopoietic necrosis virus) using the EZEE-GENE® nested PCR test kit (BIOTEC, Pathum Thani, Thailand). Shrimp were acclimatized in a 600-L tank and maintained at a salinity of 30 ppm and temperature of 25–32˚C at the Aquaculture Product Development and Services Laboratory, BIOTEC, Thailand, for the duration of the study. Shrimp were fed twice a day with StarFeed commercial pellets (CPF, Bangkok, Thailand). Water quality in the rearing tank was evaluated every three days by measuring

the temperature, pH and dissolved oxygen levels. Ammonia–nitrogen, nitrite–nitrogen and alkalinity levels were monitored weekly.

To determine the effects of ibuprofen on prostaglandin levels in haemolymph, shrimp were intramuscularly injected with PBS (control), or 40 μg or 400 μg ibuprofen. At 48 h after the injection, haemolymph were drawn, mixed with 10% ice-cold sodium citrate (anti-coagulant) at 1:1 (v/v) and centrifuged at 1,500 ×g for 5 min at 4°C to remove haemocytes. Levels of $PGE_2$ in haemolymph were then estimated by EIA. Percentage of PvCOX inhibition under inhibitor treatment was determined using the same equation as shown for the *in vitro* assay.

## Results

A functional prostaglandin biosynthesis pathway was proposed in *P. monodon* based on the detection of three prostaglandins, namely $PGD_2$, $PGE_2$ and $PGF_{2\alpha}$, in shrimp intestines with UPLC-HRMS/MS (S1 Data) and the identification of nine full-length prostaglandin biosynthesis genes (S2 Table). Among these nine genes, *Pmcox* was selected for further characterization as it controls the rate-limiting step into the prostaglandin biosynthesis pathway.

### Sequence analysis and domain prediction of PmCOX

The full-length *Pmcox* cDNA is 2,816 bp in length, containing a 1,842-bp coding region (NCBI accession number KF501342) that translates to a polypeptide 614 amino acids in length. The 5′ and 3′ UTRs are 205- and 766-bp long, respectively. TBLASTX analysis revealed that the closest PmCOX homolog is *P. vannamei* COX (PvCOX, GenBank accession no. XP027218437, E-value 0.0) with 94.95% sequence identity. CDART algorithm predicted that the putative PmCOX enzyme contains three conserved domains, which are the calcium-binding epidermal growth factor-like domain (residues 57–95), the prostaglandin endoperoxide synthase domain (residues 115–602) and the peroxidase domain (residues 172–600) (Fig 1A). Multiple sequence alignment revealed that PmCOX contains all seven essential catalytic residues, namely R146, H233, Y383, Y413, H414, H416 and S556, which were highly conserved with those in mammalian homologs (Fig 1B). Additionally, these conserved residues were also identified in COX sequences of *P. vannamei*, the kuruma prawn *P. japonicus*, the blue crab *Callinectes sapidus*, the American lobster *Homarus americanus*, the Hawaiian red shrimp *Halocaridina rubra* and the two amphipod crustaceans *Gammarus* sp. and *Caprella* sp. Interestingly, although mammalian COXs typically contain the KDEL/STEL-type endoplasmic reticulum retention/retrieval signal, the C-termini of PmCOX and crustacean COXs are shorter than the mammalian homologs and lack these signals.

AU-rich elements (AREs; AUUUA) and potential polyadenylation AAUAAA signals have previously been identified in the 3′ UTR region of COX from two amphipod crustaceans, *Gammarus* sp. and *Caprella* sp. In the *Pmcox* gene, the 3′ UTR also contains five AREs and one polyadenylation signal (Fig 1C). The presence of conserved domains and residues in both the coding region and the 3′ UTR suggested that PmCOX could likely regulate the production of $PGH_2$ similar to COXs in other organisms.

### Identification of *N*-glycosylation sites on PmCOX

For mammalian COXs to become catalytically active, three asparagine residues in COX1 and up to four asparagine residues in COX2 must be glycosylated. Multiple sequence alignment matched two known glycosylation sites in ovine COX1, N68 and N144, with PmCOX N93 and N170, respectively (Fig 1B, blue highlight). However, the third glycosylation site in ovine COX1 at N410 did not align with any asparagine residues from PmCOX or COXs from other crustaceans. As a result, the PmCOX sequence was therefore scanned for glycosylation motifs

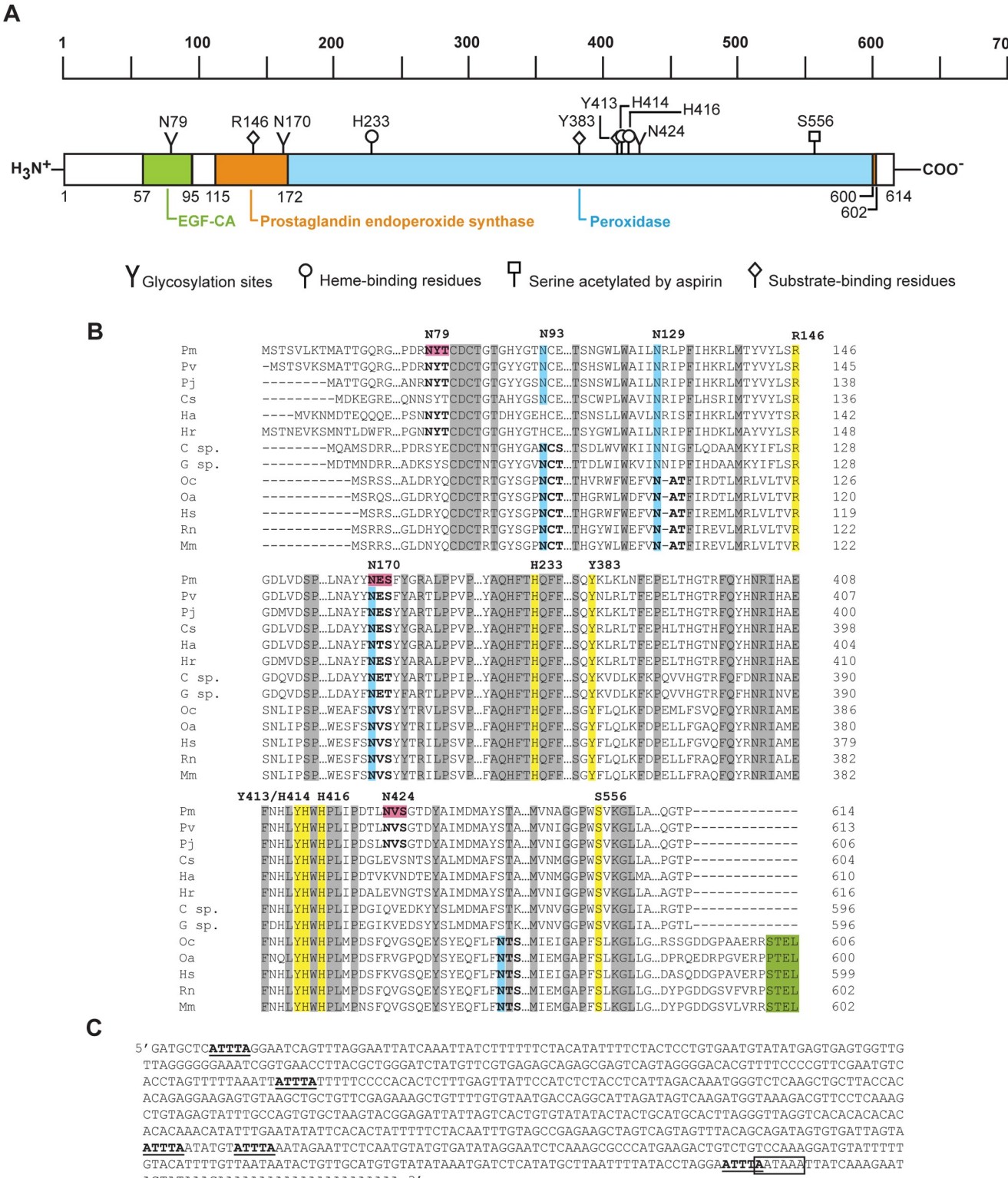

**Fig 1. Sequence analysis of PmCOX.** A) CDART analysis of PmCOX revealed the presence of a calcium-binding epidermal growth factor-like domain (EGF-CA, green box), a prostaglandin endoperoxide synthase domain (orange box) and a peroxidase domain (blue box). Predicted heme-binding residues, substrate-binding residues and an aspirin acetylation site were identified using multiple sequence alignment. *N*-glycosylation sites were predicted using NetNGlyc 1.0. B) Multiple sequence alignment of PmCOX and its homologs revealed the presence of seven conserved catalytic residues (yellow highlights). Four glycosylation sites were present only in mammalian homologs (blue highlights). Glycosylation motifs (N-(X)-S/T) are shown in bold, with PmCOX

motifs at residues 79, 170 and 424 predicted by NetNGlyc 1.0 (pink highlights). The endoplasmic reticulum retention/retrieval signal (KDEL/STEL motif, green highlights) was conserved only among the four mammalian COX sequences. Genus and species used in this alignment are abbreviated as follows: Pm–*P. monodon*, Pv–*P. vannamei*, Pj–*P. japonicus*, Cs–*C. sapidus*, Ha–*H. americanus*, Hr–*H. rubra*, C sp.–*Caprella* sp., G sp.–*Gammarus* sp., Oc–*Oryctolagus cuniculus*, Oa–*Ovis aries*, Hs–*Homo sapiens*, Rn–*Rattus norvegicus* and Mm–*Mus musculus*. C) Sequence analysis of the 3′ UTR of *Pmcox* cDNA. Five AU-rich elements (AREs) "AUUUA", or "ATTTA" for the cDNA sequence (underlined), were identified along with a single potential mRNA instability motif "AAUAAA", or "AATAAA" for the cDNA sequence (boxed), which overlaps the 3′ end of the fifth ARE.

(N-(X)-S/T), and three potential glycosylation sites were identified at N79, N170 and N424 (Fig 1B, pink highlight). The presence of these motifs suggested that the *N*-glycosylation pattern may be conserved in crustacean COXs.

## Phylogenetic analysis of PmCOX

A phylogenetic tree of COX sequences was constructed using the neighbour-joining method, revealing two distinct clusters of vertebrate and crustacean COXs. Vertebrate COXs were further separated into two major branches, corresponding to COX1 and COX2 isoforms. COX sequences from penaeid shrimp, including *P. monodon*, *P. vannamei* and *P. japonicus*, clustered together (Fig 2) and also formed part of a larger cluster that includes COX sequences from other decapod crustaceans, including *C. sapidus*, *H. americanus* and *H. rubra*. COXs from two amphipod crustaceans, *Gammarus* sp. and *Caprella* sp., were part of a separate arm from the decapod crustacean cluster. Lastly, the *D. pulex* COX sequence was the most evolutionarily distant from the rest of the group.

## PmCOX expression increased levels of PGE$_2$ in 293T cells

To determine PmCOX enzymatic function, 293T cells were untreated (control 1), transfected with pEGFP-N1 (control 2), or transfected with pcDNA3.1$^{TM}$(+)*Pmcox*/*myc*-His B. Forty-eight hours after transfection, cells were incubated with either 0.1% DMSO (vehicle) or 10 μM ARA for 10 min at 37°C. Transfected cells and cell culture media were collected separately for Western blot analysis and EIA, respectively. It was predicted that the PmCOX-Myc-His molecular mass would be approximately 70 kDa. However, Western blot analysis revealed that PmCOX expressed in 293T cells had a higher molecular weight than expected, at 77 kDa (Fig 3A), suggesting the presence of post-translational modifications on the PmCOX protein.

EIA was performed to determine the effects of ARA treatment on PmCOX-expressing cells. In the DMSO-treated group, levels of secreted PGE$_2$ were 0.23, 0.22 and 1.20 nM in cell culture media collected from control 1, control 2 and cells expressing PmCOX-Myc-His, respectively (Fig 3B). As PmCOX expression increased levels of secreted PGE$_2$ by 5.2- to 5.6-fold compared to the control groups, these findings indicated that PmCOX possessed similar enzymatic function as mammalian COX enzymes and was incorporated as part of the prostaglandin biosynthesis pathway in 293T cells.

In ARA-treated samples, levels of secreted PGE$_2$ were 1.19 and 1.21 nM for controls 1 and 2, respectively. For PmCOX-expressing cells treated with ARA, 72.54 nM PGE$_2$ was detected, which was approximately 60-fold higher than in both controls 1 and 2. These findings further confirmed the function of PmCOX. Additionally, the effects of ARA treatment were more pronounced in PmCOX-expressing samples, with PGE$_2$ levels from PmCOX-expressing cells increasing by 60-fold after ARA treatment compared to around 5-fold for controls 1 and 2.

## UPLC-HRMS/MS analysis of PGE$_2$, PGF$_{2\alpha}$ and ARA in cell culture media

To verify EIA results, 293T cells were transfected with either pcDNA3.1$^{TM}$ (+)/*myc*-His B (empty vector) or pcDNA3.1$^{TM}$ (+)*Pmcox*/*myc*-His B. Transfected cells were treated with 10,

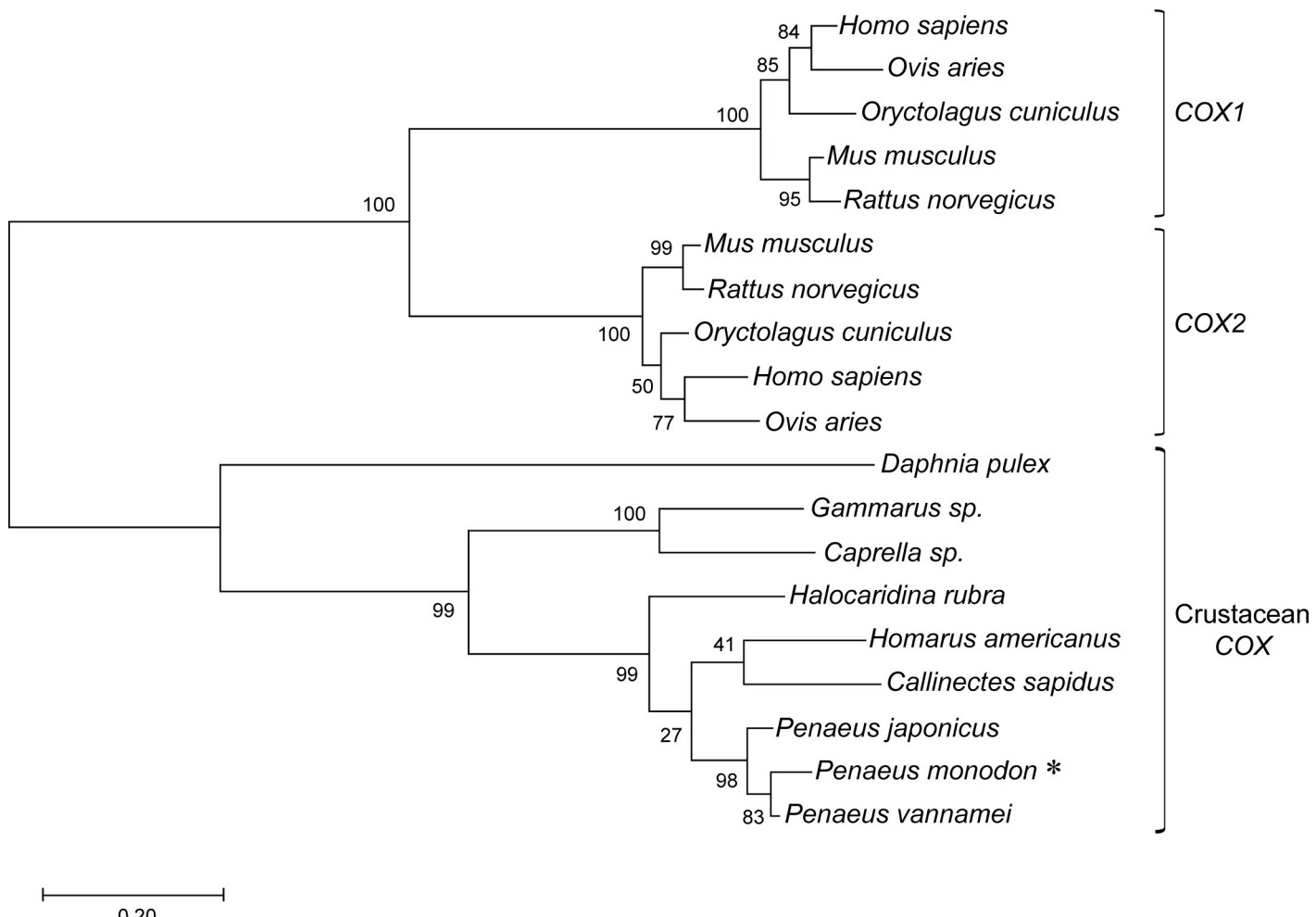

**Fig 2. Phylogenetic tree based on PmCOX sequence and its homologs.** Primary amino acid sequences of PmCOX (asterisk) and COX homologs were obtained from GenBank. Sequence accession numbers are provided in S3 Table. A phylogenetic tree was constructed using MEGA version 10.1 with the maximum likelihood program. Numbers at the nodes indicate bootstrap values from the neighbour-joining analysis. The bar labeled 0.20 represents sequence divergence.

20 or 40 μM ARA or 0.1% DMSO (vehicle) for 10 min at 37°C. Cell culture media were harvested, subjected to C18 SPE and analyzed by UPLC-HRMS/MS. The base peak chromatogram of cell culture extracts showed peaks at retention times 5.36, 5.84 and 16.61 min, which corresponded with the retention times of $PGF_{2\alpha}$, $PGE_2$ and ARA standards, respectively (Fig 4A). The mass spectra and product ions of $PGF_{2\alpha}$, $PGE_2$ and ARA were used to verify identities of these metabolites (Fig 4B–4D). Levels of $PGF_{2\alpha}$ and $PGE_2$ were determined using the standard curve method. On the other hand, levels of ARA in cell culture medium exceeded the linearity range and were not quantified in this analysis. To establish prostaglandin basal levels, non-transfected controls and cells expressing PmCOX were incubated with 0.1% DMSO as negative controls. UPLC-HRMS/MS analyses revealed that levels of $PGE_2$ and $PGF_{2\alpha}$ were below the detection limit when cells were treated with 0.1% DMSO, suggesting that basal levels of these prostaglandins were negligible in this assay (Fig 4E). When non-transfected controls were incubated with 40 μM ARA, 1.07 nM $PGE_2$ was detected in cell culture media while $PGF_{2\alpha}$ remained under the limit of detection. The production of $PGE_2$ and $PGF_{2\alpha}$ was more robust only in PmCOX-expressing cells that were treated with ARA. In fact, levels of secreted

**A**

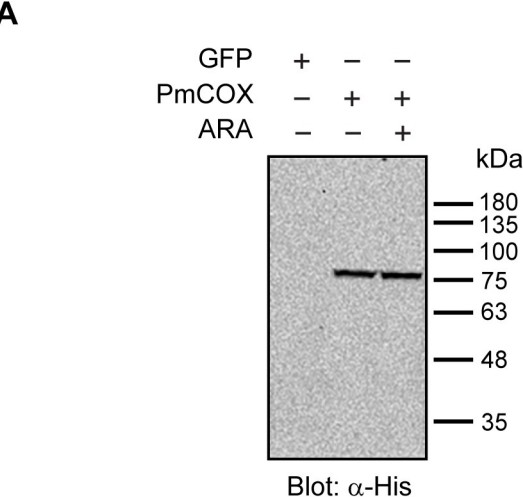

**B**

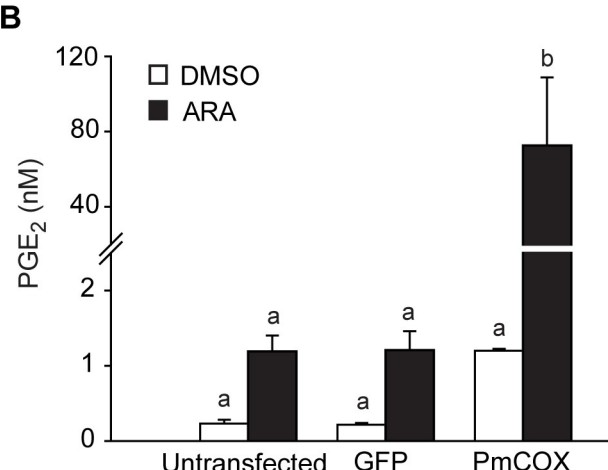

**Fig 3. PmCOX enzymatic activity was determined via protein expression in mammalian cells.** Cells were transiently transfected with pEGFP-N1 (control 1), pcDNA3.1$^{TM}$ (+)*Pmcox*/*myc*-His B or left untransfected (control 2). Forty-eight hours after transfection, cells were incubated with 0.1% DMSO (vehicle) or 10 μM ARA for 30 min. (A) Western blot analysis was performed using transfected cell lysates to verify the expression of PmCOX-Myc-His using anti-His antibodies. (B) Levels of secreted PGE$_2$ in cell culture media were estimated by EIA. Error bars show standard deviations. Different letters indicate statistically significant differences in PGE$_2$ levels as determined by Duncan's test. The *p*-value was 0.000 for this analysis.

PGF$_{2\alpha}$ and PGE$_2$ corresponded with levels of exogenous ARA used in the treatment in a dose-dependent manner. Incubation with 10, 20 and 40 nM ARA resulted in 0.90, 1.08 and 1.65 nM PGE$_2$ and 0.21, 0.21 and 0.36 nM PGF$_{2\alpha}$, respectively.

### *N*-glycosylation of PmCOX

The presence of three *N*-glycosylation motifs in the PmCOX protein sequence (Fig 1B) suggested that *N*-glycosylation is required for PmCOX to be catalytically active. Pulled-down PmCOX-Myc-His was incubated with endoglycosidase H, which cleaves oligosaccharides from asparagine residues of glycoproteins, resulting in decreased protein molecular mass (Fig 5A). Interestingly, the sizes of glycosylated PmCOX and endoglycosidase H-treated PmCOX in this assay were slightly higher than our other Western blot analyses at approximately 90 and

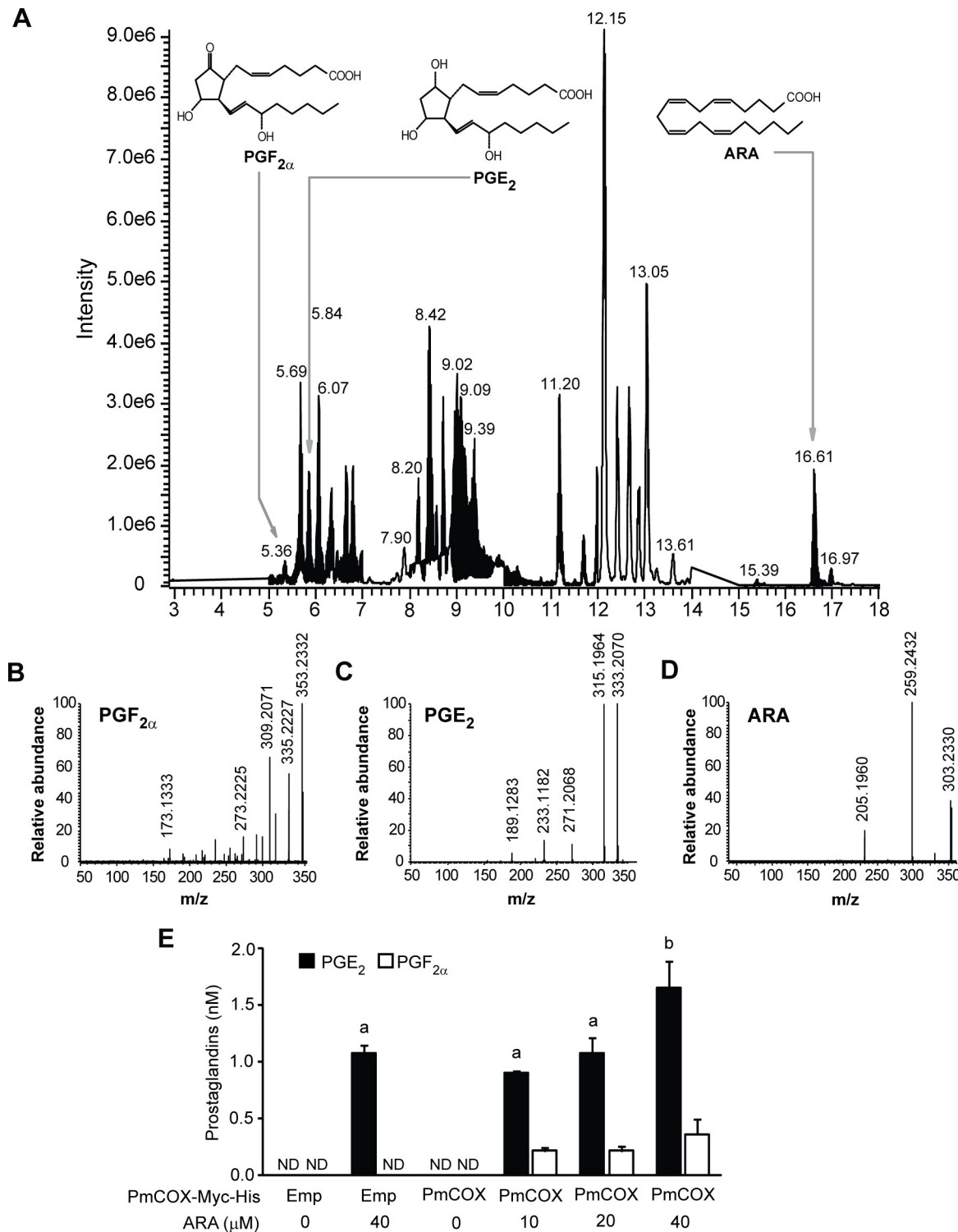

**Fig 4. Analysis of PGF$_{2\alpha}$, PGE$_2$ and ARA in cell culture media by UPLC-HRMS/MS.** Cells were transiently transfected with either pcDNA3.1$^{TM}$ (+)/*myc*-His B (empty vector) or pcDNA3.1$^{TM}$ (+)*Pmcox*/*myc*-His B. Forty-eight hours after transfection, cells were incubated with 10, 20 or 40 μM ARA at 37°C for 10 min. Cell culture media were harvested, subjected to C18 SPE and analyzed by UPLC-HRMS/MS. (A) The base peak chromatogram revealed the mass spectra of PGF$_{2\alpha}$, PGE$_2$ and ARA in cell culture media of 293T cells expressing PmCOX-Myc-His. Extracted mass profiles of (B) PGF$_{2\alpha}$, (C) PGE$_2$ and (D) ARA were selected by matching the retention times to those of the commercially available standards at 5.36, 5.84 and 16.61 min, respectively. (E) Levels of PGE$_2$ (black bar) and PGF$_{2\alpha}$ (white bar) secreted by cells either transfected with the empty vector (Emp) or expressing PmCOX-Myc-His proteins while being treated with 0, 10, 20 or 40 nM ARA. The *p*-value for PGE$_2$ was 0.001.

80 kDa respectively, possibly due to the effects of the endoglycosidase H digestion mixture on protein migration.

To confirm the *N*-glycosylation of PmCOX, cells expressing PmCOX-Myc-His were treated with tunicamycin, which inhibits *N*-linked glycosylation by preventing the addition of oligo-saccharide to nascent polypeptides. Western blot analysis revealed that the tunicamycin-treated sample resulted in two protein bands at high (approximately 77 kDa) and low (approximately 70 kDa) molecular weights, which corresponded with glycosylated and unglycosylated PmCOX, respectively (Fig 5B). Based on these results, it could be concluded that PmCOX was expressed in 293T cells as an *N*-glycosylated protein with a molecular mass of 77 kDa.

## Mutational analysis to determine PmCOX glycosylation sites

Site-directed mutagenesis was performed to disrupt PmCOX potential glycosylation sites at residues 79, 170 and 424 to create three PmCOX single glycosylation mutants (PmCOXN79Q, PmCOXN170Q and PmCOXN424Q), three double glycosylation mutants (PmCOXN79QN170Q, PmCOXN170QN424Q and PmCOXN79QN424Q), and one triple glycosylation mutant (PmCOXN79QN170QN424Q). Western blot analysis revealed that PmCOX molecular mass gradually decreased as the number of glycosylation mutations increased (Fig 5C), suggesting that all three asparagine residues were glycosylated at the same time when expressed in mammalian cells.

## Enzymatic activities of PmCOX glycosylation mutants were determined by EIA and UPLC-HRMS/MS

To determine whether *N*-glycosylation is required for PmCOX catalytic function, cells trans-fected with the empty vector or vectors encoding wild-type PmCOX, PmCOX single glycosyla-tion mutants or the PmCOX triple glycosylation mutant were treated with either 0.1% DMSO or 40 μM ARA at 37°C for 30 min. Western blot analysis was performed to verify protein expression and to demonstrate the differences in molecular mass for wild-type PmCOX and various PmCOX glycosylation mutants (Fig 6A).

Due to the presence of endogenous COX in 293T cells, the basal level of $PGE_2$ was deter-mined using cell culture media collected from 293T cells transfected with the empty vector. EIA revealed that levels of $PGE_2$ were 0.64 and 2.68 nM when treated with 0.1% DMSO and 40 μM ARA, respectively (Fig 6B). This indicated that endogenous COX enzyme in 293T cells was able to convert exogenous ARA into secreted $PGE_2$ in cell culture media. On the other hand, when wild-type PmCOX was expressed in 293T cells without ARA treatment, $PGE_2$ lev-els were only 0.62 nM, comparable to the basal level of $PGE_2$ in cells transfected with the empty vector. Therefore, the availability of ARA as a COX substrate had a larger impact on lev-els of secreted $PGE_2$ than the amount of COX enzyme present in 293T cells.

To determine whether mutations at the PmCOX glycosylation sites resulted in reduced PmCOX catalytic function, cells expressing wild-type PmCOX or PmCOX glycosylation mutants were treated with 40 μM ARA. Expression of wild-type PmCOX resulted in 7.40 nM $PGE_2$ (Fig 6B). Expression of PmCOXN79Q, PmCOX170Q, and PmCOXN424Q resulted in 8.65, 2.45 and 2.79 nM of secreted $PGE_2$, respectively. This suggests that *N*-glycosylation at res-idues 170 and 424, but not residue 79, is required for PmCOX catalytic function. Additionally, expression of the PmCOX triple mutant resulted in 2.23 nM secreted $PGE_2$. As levels of $PGE_2$ from PmCOXN170Q, PmCOXN424Q and the PmCOX triple mutant were comparable to those found in cells transfected with the empty vector, it was postulated that the removal of either N170 or N242 glycosylation site is sufficient to completely disrupt the PmCOX enzy-matic activity.

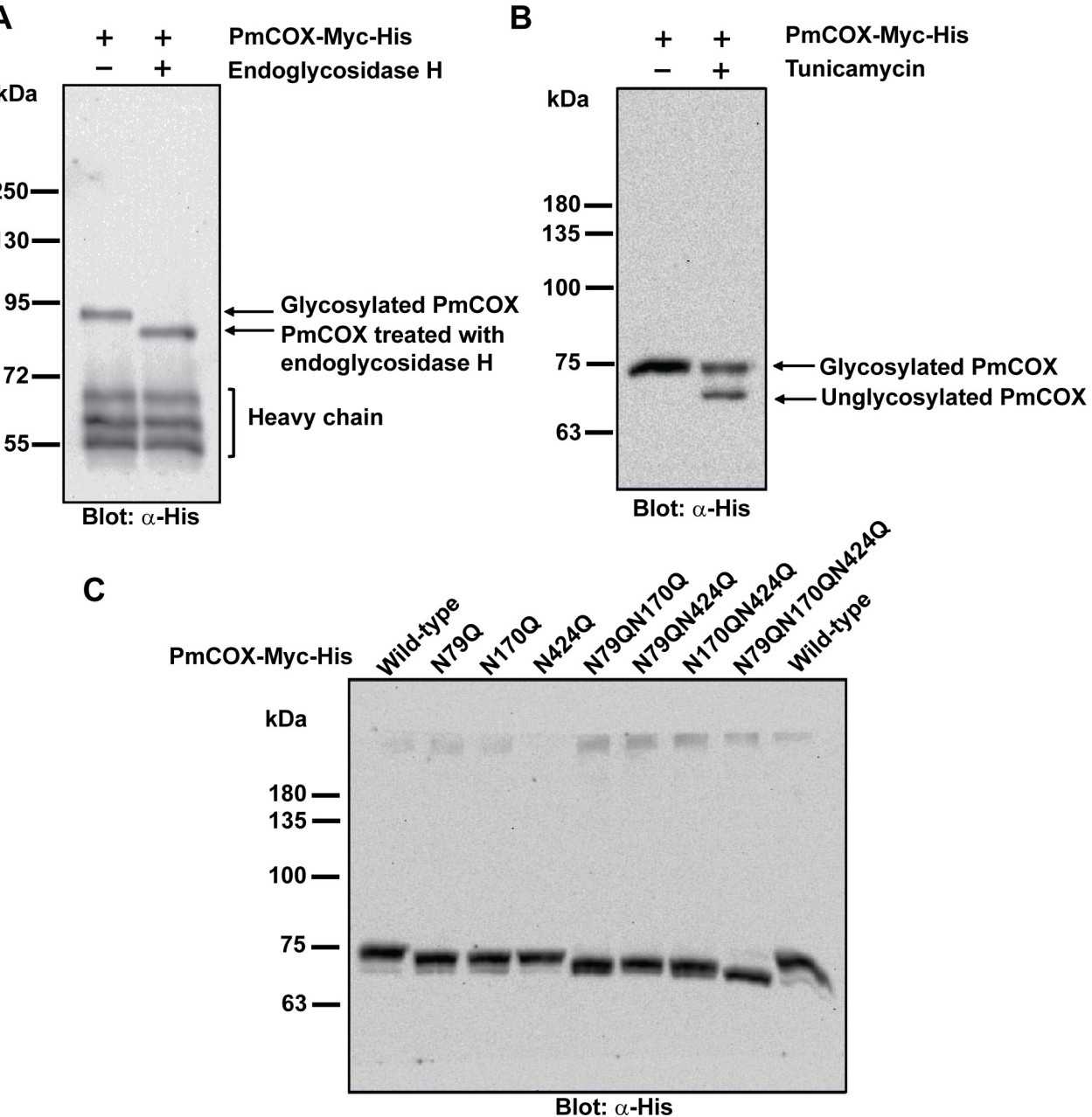

**Fig 5. PmCOX *N*-glycosylation patterns.** (A) Pulled-down PmCOX-Myc-His was denatured at 95˚C for 5 min and incubated with the digestion mixture with or without endoglycosidase H for 18 h at 37˚C. Western blot analysis was performed to compare the molecular mass of the PmCOX-Myc-His with and without endoglycosidase H treatment. (B) Cells expressing PmCOX-Myc-His were treated with 2 µg/mL tunicamycin or 0.1% DMSO (vehicle) for 16 h at 37˚C. Cells were harvested and Western blot analysis was performed to determine changes in the molecular mass of the PmCOX protein. (C) Western blot analysis was performed to determine the molecular mass of wild-type PmCOX and PmCOX with single, double and triple glycosylation mutants.

UPLC-HRMS/MS analysis was performed on cell culture medium harvested from cells expressing wild-type or triple-mutant PmCOX in the presence or absence of exogenous ARA. In the absence of ARA, levels of $PGE_2$ and $PGF_{2\alpha}$ were below the detection limit regardless of whether cells were transfected with the empty vector or expressing wild-type or triple-mutant PmCOX (Fig 6C). In the presence of 40 µM ARA, however, levels of secreted $PGE_2$ were 1.1,

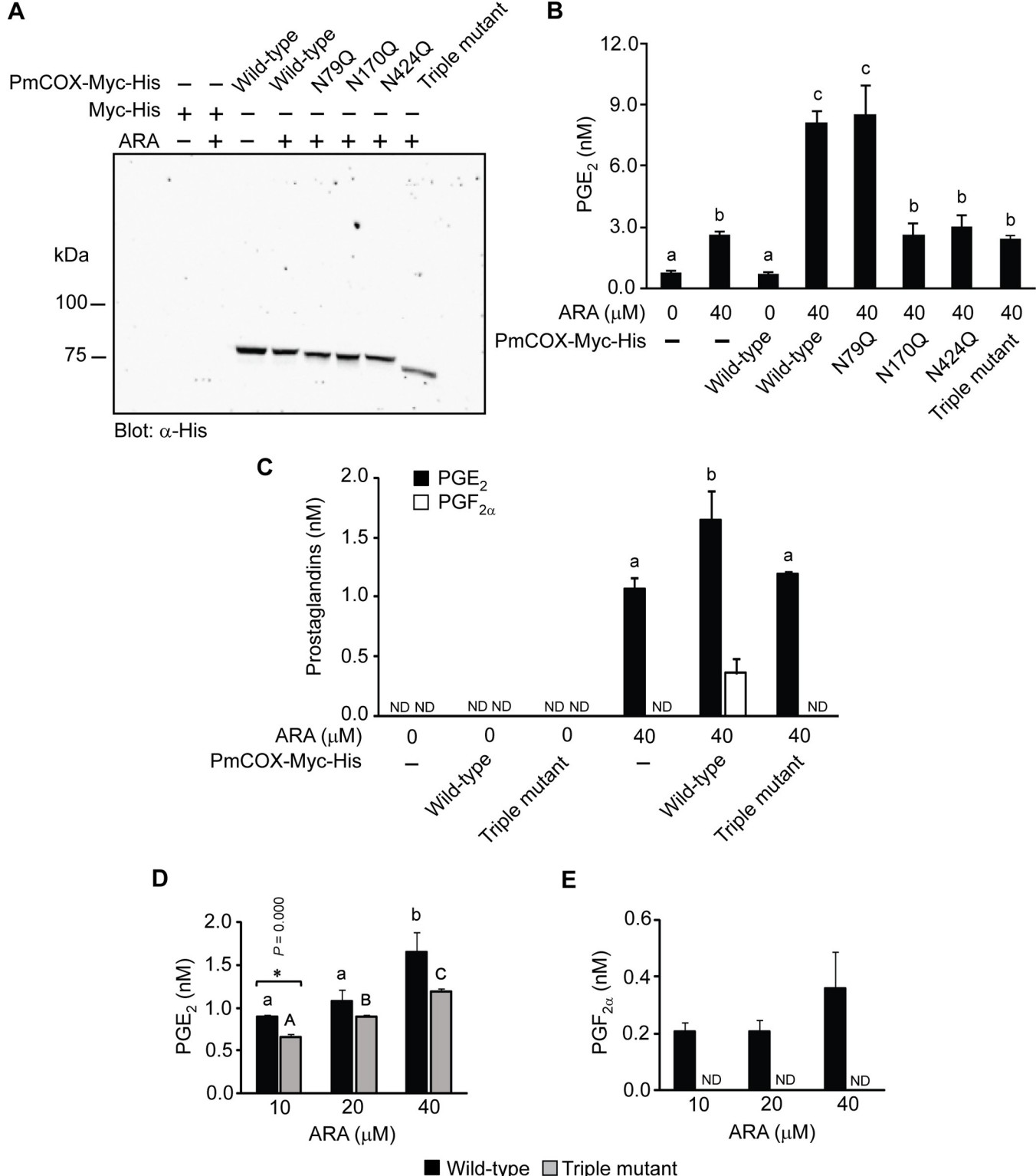

**Fig 6. Effects of *N*-glycosylation on PmCOX enzymatic activities.** Wild-type PmCOX and PmCOX single and triple glycosylation mutants were expressed in 293T cells. Forty-eight hours after transfection, cells were incubated with either 0.1% DMSO or 40 μM ARA at 37˚C for 10 min. Transfected cells and cell culture media were harvested to verify PmCOX protein expression and levels of prostaglandins, respectively. (A) Western blot analysis was performed to confirm protein expression. (B) Levels of PGE$_2$ in cell culture media were estimated by EIA. (C) The remaining culture media were extracted by C18 SPE and analyzed by UPLC-HRMS/MS to determine levels of PGE$_2$ and PGF$_{2\alpha}$. To determine the effects of substrates, cells expressing wild-type PmCOX or the

PmCOX triple glycosylation mutant were treated with 10, 20 or 40 μM ARA for 30 min at 37˚C. Levels of (D) $PGE_2$ and (E) $PGF_{2\alpha}$ were determined by UPLC-HRMS/MS. Error bars represent standard deviations. ND indicates that the level of prostaglandin was below the detection limit of UPLC-HRMS/MS. Different letters indicate statistically significant differences in $PGE_2$ levels as determined by the Duncan test ($p = 0.000$ in Fig 6B; $p = 0.007$ in Fig 6C; $p = 0.003$ and 0.000 for wild-type and mutant COX, respectively, in Fig 6D). Asterisks (*) indicate a significant difference between levels of secreted $PGE_2$ between wild-type PmCOX and the PmCOX triple glycosylation mutant using the t-test ($p = 0.000$).

1.7 and 1.2 nM from cells transfected with the empty vector and cells expressing wild-type or triple-mutant PmCOX, respectively (Fig 6C, black bars). On the other hand, $PGF_{2\alpha}$ was detected only in cells expressing wild-type PmCOX but not in cells transfected with the empty vector or those expressing the PmCOX triple mutant (Fig 6C, white bars). These findings confirmed that the removal of *N*-glycosylation sites on PmCOX disrupted the production of $PGE_2$ and $PGF_{2\alpha}$ in 293T cells.

To investigate whether the effects of PmCOX *N*-glycosylation mutations were observed only when cells were treated with high concentrations of ARA, cells expressing wild-type or triple-mutant PmCOX were treated with 10, 20 or 40 nM ARA. UPLC-HRMS/MS analyses revealed that levels of $PGE_2$ steadily increased in a dose-dependent manner (Fig 6D). However, $PGF_{2\alpha}$ levels were mostly unaffected, with comparable amounts being secreted from cells expressing wild-type PmCOX while being undetectable from cells expressing the PmCOX triple mutant (Fig 6E).

### *In vitro* and *in vivo* COX inhibition assays in haemolymph of penaeid shrimp

Although our data indicated that PmCOX was catalytically active, PmCOX enzymatic function had yet to be demonstrated in *P. monodon*. Based on conserved catalytic residues and domains between PmCOX and its mammalian homologs, we hypothesized that commercially available COX inhibitors could reduce PmCOX catalytic function in shrimp. *In vitro* inhibition assays were performed using *P. monodon* haemolymph. Haemolymph treated with 10 μM ARA (positive control) was set as 0% inhibition of the PmCOX enzymatic activity (Fig 7A). Haemolymph samples were also pre-treated with varying concentrations of aspirin, ibuprofen or indomethacin for 30 min at 28˚C followed by 10 μM ARA for an additional 1 h. Incubation with 150 ng/mL aspirin and 200 ng/mL ibuprofen resulted in the highest and second highest inhibitory effects at 62.3% and 57.0% inhibition, respectively. Based on the ability of aspirin and ibuprofen to reduce $PGE_2$ levels in shrimp haemolymph, we propose that a functional PmCOX is present in *P. monodon*.

Based on the *in vitro* inhibition assays where aspirin and ibuprofen effectively reduced $PGE_2$ levels in *P. monodon* haemolymph, the effects of COX inhibitors were subsequently verified *in vivo*. As aspirin is poorly soluble in water, ibuprofen was selected for intramuscular injection in shrimp. Although we intended to repeat the experiment in *P. monodon*, specific pathogen free (SPF) *P. monodon* of appropriate size could not be obtained for the experiment. *P. vannamei* was selected as a substitute based on the presence of prostaglandin biosynthesis genes (S2 Table) and the identification of $PGE_2$ and $PGF_{2\alpha}$ in *P. vannamei* post-larvae (S6 Data). Therefore, SPF *P. vannamei* was chosen for testing the effects of ibuprofen *in vivo*. *P. vannamei* were intramuscularly injected with 100 μL PBS (vehicle) or ibuprofen at a dose of 40 or 400 ng per gram shrimp. As *P. vannamei* body weights were approximately 12 g per shrimp, 0.48 or 4.8 μg ibuprofen were injected into each shrimp. Levels of $PGE_2$ in haemolymph of *P. vannamei* injected with PBS were 933.5 pM, which was set at 0% inhibition (Fig 7B). Levels of $PGE_2$ in haemolymph of *P. vannamei* injected with 0.48 and 4.8 μg of ibuprofen were 93.1 and 124.3 pM, which were equivalent to 90.0% and 86.7% inhibition, respectively. We speculated

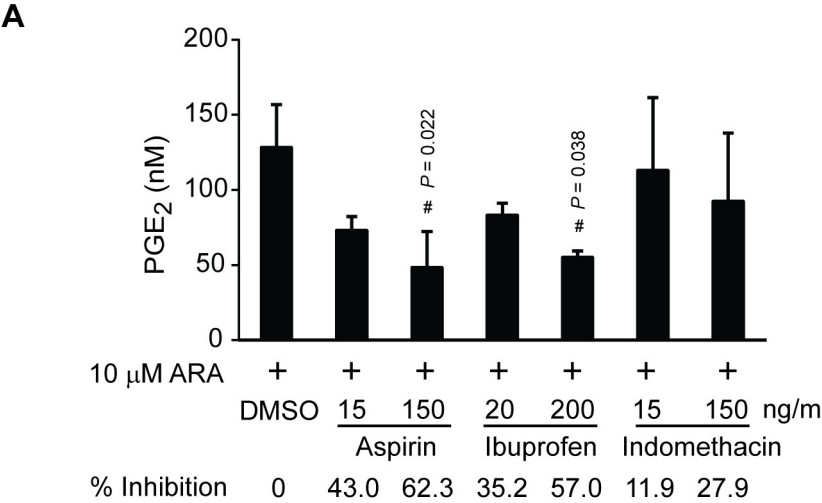

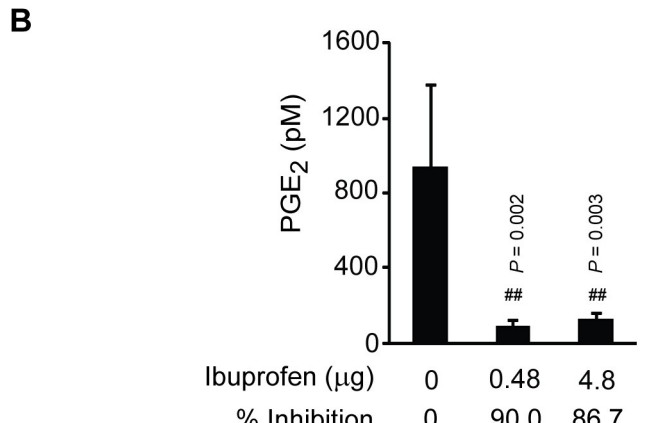

**Fig 7. Effects of COX inhibitors on levels of PGE₂ in shrimp haemolymph.** (A) *In vitro* inhibition assays were performed using *P. monodon* haemolymph. Haemolymph samples were incubated with 0.1% DMSO, 15 or 150 ng/mL aspirin, 20 or 200 ng/mL ibuprofen, or 15 or 150 ng/mL indomethacin at 28°C for 30 min. The mixtures were then incubated with 10 μM ARA at 28°C for 30 min. Levels of PGE₂ in haemolymph samples were estimated by EIA. Response of PmCOX to each inhibitor is shown as % inhibition. The experiment was performed in triplicate (*n* = 3). (B) *In vivo* inhibition assays were performed by intramuscular injection of 100 μL PBS or 0.48 or 4.8 μg ibuprofen into *P. vannamei* (*n* = 4 per treatment). Shrimp were maintained in separate glass tanks for 48 h post-injection. EIA was used to determine levels of PGE₂ in haemolymph. Pound sign (#) indicates a significant difference between levels of PGE₂ in designated samples compared to the control group using Dunnett's test.

that PGE₂ levels in shrimp haemolymph were not reduced in a dose-dependent manner when treated with ibuprofen because the ibuprofen dosage at 40 ng/g shrimp already exceeded the highest concentration required to inhibit the function of *P. vannamei* COX (PvCOX) *in vivo*. Nevertheless, results from both *in vitro* and *in vivo* inhibitory assays confirmed the presence of enzymatically active COX enzyme in two penaeid shrimp.

## Discussion

The prostaglandin biosynthesis pathway has been shown to regulate crustacean reproductive maturation, especially in economically important species such as *P. monodon*, the crab *Oziothelphusa senex senex*, *M. japonicus* and *M. rosenbergii* [24, 27, 38, 41–44]. However, the

presence of functional COX enzymes in these decapod crustaceans has yet to be confirmed. In this study, sequence analysis revealed that PmCOX and other crustacean COX sequences contain conserved catalytic residues and domains essential for their catalytic function. The function of PmCOX was confirmed as protein expression in mammalian cells resulted in increasing levels of secreted $PGE_2$ and $PGF_{2\alpha}$. Similar to mammalian COX, PmCOX contains *N*-glycosylation sites at residues N79, N170 and N424. The removal of glycosylation sites at either residue 170 or 424 completely abolished PmCOX enzymatic activity, suggesting that PmCOX requires *N*-glycosylation to be catalytically active. Lastly, COX enzymatic activities in *P. monodon* and *P. vannamei* were demonstrated in *in vitro* and *in vivo* inhibition assays respectively, suggesting the conserved nature of COX enzymes among penaeid shrimp.

## Conserved catalytic residues in COX active sites

One of the strategies used to identify candidate COX enzymes, such as those in the fruit fly *Drosophila melanogaster*, is the mapping of essential catalytic residues in the COX active site [45, 46]. These catalytic residues were identified by comparing with those previously characterized in ovine and human COXs [47, 48]. For example, residues R120 and Y355 in ovine COX1 (OvCOX1), which bind the carboxylic acid moiety of ARA, are equivalent to residues R146 and Y383 in PmCOX, respectively [49–51]. Similarly, OvCOX1 Y385 is responsible for extracting a hydrogen from ARA and is equivalent to PmCOX Y413 [47], a highly conserved residue in all crustacean COXs. Distal and proximal histidines, which are responsible for hydroperoxide reductase activities and coordinating the heme iron inside COX active site, are equivalent to residues H207 and H338 in OvCOX1 and residues H233 and H414 in PmCOX, respectively [16, 52, 53]. Lastly, OvCOX1 S530, which serves as the acetylation site for aspirin, aligns with S556 in PmCOX and COX enzymes in other crustaceans [47]. Our sequence alignment not only confirmed the identity of PmCOX by matching its essential catalytic residues to those in mammalian homologs, but also showed that these residues were well-conserved in crustacean COX sequences, suggesting the presence of functional COX enzymes in all analyzed crustacean species.

## Expression of invertebrate COX enzymes

Various cell-based expression systems have previously been used to establish COX enzymatic activity and *N*-glycosylation requirements, including yeast, baculovirus-based expression in insect cells and mammalian cells [54–56]. Insect SF9 cells were used for expression of mouse COX2 to investigate its *N*-glycosylation sites [54]. *Pichia pastoris* has also been used to determine the *N*-glycosylation sites of human COX1 and COX2 [55]. Mammalian COS-7 cells have been used to express COX enzymes from zebrafish and amphipod crustaceans [23, 56]. Similarly, COS-1 cells were used to determine glycosylation sites and effects of COX inhibitors on human COX1 [18, 57]. In this study, SF9 and SF21cells had been considered as possible expression hosts for PmCOX as insect cells are more closely related to shrimp. However, Shimokawa and Smith (1992) reported that the insect cell system was not suitable for expressing large amounts of COX enzyme as it lacked the ability to completely glycosylate the expressed proteins [14]. Furthermore, the characterization of COXs from *Gammarus* sp. and *Caprella* sp. was performed in COS-7 cells, serving as an example of a successful mammalian cell expression system for crustacean COX expression and characterization [23].

## *N*-glycosylation requirement

Aside from conserved catalytic residues, another important characteristic of the COX enzyme is its *N*-glycosylation requirement. Site-directed mutagenesis was used to identify glycosylation

residues of COX1 and COX2 [19], and revealed that COX1 is glycosylated at three asparagine residues (N68, N144 and N410) while COX2 may be glycosylated at up to four asparagine residues with a fourth glycosylation site at N580 [19, 53, 54]. It has been demonstrated that the first three glycosylation sites on COX1 are required for protein expression and enzymatic function [19]. On the other hand, COX2 glycosylation at N580 affects COX2 dimer formation, protein stability and efficacy of certain COX inhibitors [18, 57, 58].

The requirement for *N*-glycosylation was also conserved in invertebrate COXs. Four and six potential glycosylation motifs were identified in *G. fruticosa* COX-A and COX-B, respectively [21]. Expression of these COX isoforms in COS-7 cells treated with tunicamycin revealed that COX-A and COX-B were glycosylated at four and five glycosylation sites, respectively [21]. Similarly, tunicamycin treatment was also used to verify *N*-glycosylation on COX enzymes from two species of amphipod crustaceans *Gammarus* sp. and *Caprella* sp. [23]. Nevertheless, whether COX *N*-glycosylation is required for its enzymatic function in invertebrates has not previously been investigated. In this study, we demonstrated that PmCOX was *N*-glycosylated at three glycosylation sites and mutations at N170 or N424 completely disrupted PmCOX enzymatic function. Furthermore, Western blot analysis revealed decreased expression levels of the triple-mutant PmCOX compared to those of the wild type or single-glycosylation mutants, suggesting that removal of more than one glycosylation site may affect PmCOX protein folding or protein turnover rates. Further investigation is required to confirm the effects of *N*-glycosylation on PmCOX protein stability.

## Effects of COX inhibitors on female reproductive maturation in crustaceans

Effects of COX inhibitors on female reproductive maturation have often been used as evidence for the presence of a prostaglandin biosynthesis pathway in crustaceans [44, 45, 66]. For example, chronic ibuprofen treatment in the small planktonic crustacean *Daphnia magna* reduced the number of offspring in a dose-dependent manner [39]. In *O. senex senex*, injection with aspirin or indomethacin reduced the ovarian index, oocyte diameters and vitellogenin levels in crab ovaries [38]. Surprisingly, periodic injection of ibuprofen at 0.1 μg/g body weight into eyestalk-ablated *P. vannamei* resulted in higher rates of females with developing ovaries compared to the control [59]. Although these studies provide compelling evidence regarding the effects of COX inhibitors on crustacean ovarian development, changes in prostaglandin levels in these organisms after COX inhibitor treatment have not been examined. Our study is the first to report changes in levels of prostaglandins after ibuprofen injection in crustaceans. However, whether the COX inhibitor would also increase ovarian maturation rates in female *P. monodon* remains to be investigated.

## COX substrate preference and feed formulation

COX substrate specificity is regulated by cavity space within the COX active site. In mammals, ARA is the preferred substrate for COX1 whereas COX2 is more flexible, binding ARA derivatives and other PUFAs, including EPA, α-linolenic acid, γ-linolenic acid and linoleic acid as well as ARA [10, 60, 61]. In fish, COX1 and COX2 enzymes have a more restrictive substrate specificity, with preference toward ARA, but not EPA and DHA [62]. As this study confirms that ARA serves as a substrate for PmCOX, the effects of dietary ARA on shrimp prostaglandin biosynthesis should be taken into consideration during the formulation of shrimp feed. Studies have shown that levels of $PGE_2$ increased during ovarian development in the Florida crayfish *Procambarus paeninsulanus*, *O. senex senex* and *P. monodon* [24, 41, 63, 64]. As studies in mammals indicated that abundant levels of EPA or DHA could have inhibitory effects on

COX catalytic function in converting ARA to $PGE_2$ [65, 66], levels of EPA and DHA should be regulated in feed for broodstock to prevent the disruption of COX catalytic function during ovarian development.

## Supporting information

**S1 Table. Analytical characteristics of $PGF_{2\alpha}$, $PGE_2$ and ARA using UPLC-HRMS/MS.**
(DOCX)

**S2 Table. List of prostaglandin biosynthesis genes in identified in *P. monodon* and *P. vannamei*.**
(DOCX)

**S3 Table. Accession number of PmCOX and COX homologs used in the construction of phylogenetic tree.**
(DOCX)

**S1 Fig. Multiple sequence alignment of vertebrate COX1 and invertebrate COXs.**
(DOCX)

**S2 Fig. Multiple sequence alignment of vertebrate COX2 and invertebrate COXs.**
(DOCX)

**S1 Data. Extracted ion chromatograms and mass spectra of $PGE_2$, $PGD_2$ and $PGF_{2\alpha}$ in *P. monodon* intestines.**
(PDF)

**S2 Data. Enzyme immunoassay analysis of $PGE_2$ levels in PmCOX expressing cells.**
(XLSX)

**S3 Data. UPLC-HRMS/MS analysis of PUFAs and prostaglandins in medium collected from PmCOX expressing cells.**
(XLSX)

**S4 Data. Enzyme immunoassay analysis of levels of $PGE_2$ in wild-type COX and COX glycosylation mutants.**
(XLSX)

**S5 Data. Enzyme immunoassay analysis of levels of $PGE_2$ in *P. monodon* haemolymph treated with COX inhibitors.**
(XLSX)

**S6 Data. Extracted ion chromatograms and mass spectra of $PGE_2$ and $PGF_{2\alpha}$ in *P. vannamei* post-larvae.**
(PDF)

**S7 Data. Enzyme immunoassay analysis of levels of $PGE_2$ in *P. vannamei* haemolymph after injection with COX inhibitors.**
(XLSX)

**S1 Raw images.**
(PDF)

## Acknowledgments

We thank Dr. Samaporn Teeravechyan for fruitful discussions.

## Author Contributions

**Conceptualization:** Suganya Yongkiettrakul, Vanicha Vichai, Wananit Wimuttisuk.

**Formal analysis:** Punsa Tobwor, Pacharawan Deenarn, Thapanee Pruksatrakul, Surasak Jiemsup, Pisut Yotbuntueng, Wananit Wimuttisuk.

**Funding acquisition:** Punsa Tobwor, Wananit Wimuttisuk.

**Investigation:** Punsa Tobwor, Pacharawan Deenarn, Thapanee Pruksatrakul, Surasak Jiemsup, Metavee Phromson, Sage Chaiyapechara, Waraporn Jangsutthivorawat, Pisut Yotbuntueng, Oliver George Hargreaves, Wananit Wimuttisuk.

**Methodology:** Punsa Tobwor, Pacharawan Deenarn, Thapanee Pruksatrakul, Surasak Jiemsup, Suganya Yongkiettrakul, Pisut Yotbuntueng, Wananit Wimuttisuk.

**Project administration:** Sage Chaiyapechara, Waraporn Jangsutthivorawat, Wananit Wimuttisuk.

**Resources:** Punsa Tobwor, Sage Chaiyapechara, Wananit Wimuttisuk.

**Software:** Punsa Tobwor, Pacharawan Deenarn, Thapanee Pruksatrakul.

**Supervision:** Vanicha Vichai, Wananit Wimuttisuk.

**Validation:** Punsa Tobwor, Pacharawan Deenarn.

**Visualization:** Punsa Tobwor, Pacharawan Deenarn.

**Writing – original draft:** Punsa Tobwor, Wananit Wimuttisuk.

**Writing – review & editing:** Punsa Tobwor, Pacharawan Deenarn, Suganya Yongkiettrakul, Vanicha Vichai, Oliver George Hargreaves, Wananit Wimuttisuk.

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
