## [Decision Letter · Decision Letter 0]

5 Feb 2021

PONE-D-20-40833

Biochemical characterization of the cyclooxygenase enzyme in penaeid shrimp

PLOS ONE

Dear Dr. Wimuttisuk,

Thank you for submitting your manuscript to PLOS ONE. After careful consideration, we feel that it has merit but does not fully meet PLOS ONE’s publication criteria as it currently stands. Therefore, we invite you to submit a revised version of the manuscript that addresses the points raised during the review process.

The manuscript has been reviewed by two experts in the field; please find their comments appended (end of this email; attachment). Based on the reviewers’ comments as well as my own reading of the manuscript, I have decided for ‘Major Revision’ as there are two particular aspects where the data provided do not fully support the conclusions of this present manuscript.

A revised version should address all issues pointed out by the reviewers and in particular:

(1) include an independent experimental result to confirm the presence of PGE2 in this shrimp (e.g. by MS analysis);

(2) include sufficient evidence or supporting data showing that PG anabolism can occur in this shrimp (e.g. by identification of relevant enzymes based on the shrimp genome).

We look forward to receiving your revised manuscript.

Kind regards,

Andreas Hofmann

Academic Editor

PLOS ONE

Journal Requirements:

Reviewers' comments:

Reviewer's Responses to Questions

**Comments to the Author**

1. Is the manuscript technically sound, and do the data support the conclusions?

Reviewer #1: Yes

Reviewer #2: Partly

2. Has the statistical analysis been performed appropriately and rigorously? 

Reviewer #1: Yes

Reviewer #2: I Don't Know

3. Have the authors made all data underlying the findings in their manuscript fully available?

Reviewer #1: Yes

Reviewer #2: Yes

4. Is the manuscript presented in an intelligible fashion and written in standard English?

Reviewer #1: Yes

Reviewer #2: Yes

5. Review Comments to the Author

Reviewer #1: Tobwar et al. describe the expression and characterization of cyclooxygenase (COX) in penaeid shrimp and identify two asparagine residues that must be glycosylated in the catalytically active enzyme. The authors provide evidence for the enzymatic function both in vitro and in vivo and highlight similarities between various species.

Overall, the work appears to be thorough and well considered. The results are of a high standard and support the conclusions. The paper is well-written and the overall conclusions are placed in the broader context of the potential benefit to the aquaculture industry should this system be leveraged in penaeid shrimp breeding.

Comments for the authors, minor issues to be addressed and typographical errors for correction are detailed in the attached document. This is an elegant piece of work that has been well reported and the authors are to be commended on the clarity of the writing and figures. I recommend publication following minor corrections.

Reviewer #2: Major points

The major problem with this manuscript is the reliance of EIA in several crucial experiments to prove that PGE2 is actually present in this crustacean. I can see no evidence that this was checked/verified with MS or an equivalent specific method. The authors have proven that there is active COX present but generation of PGs requires other enzymes that convert the intermediates generated by COX to PGs. Without such evidence this work CANNOT be published and the authors must carry out more experiments. Finally, the lack of dose response using ibuprofen is worrying and points to side effects of this inhibitor when at too high concentration. Perhaps some lower doses of inhibitor should be tried.

Minor points

Lines 23-4. This is not accurate description of the current status of COX identity in crustaceans (see lines 291 onwards for examples of COX already described in various crustaceans)

Line 29 The role of tunicamycin needs clarifying for the reader

Lines 69-70 This is irrelevant, so remove

Lines 150-155: This is a problem as EIA is not 100% specific in terms of antibody binding. Other PGs will cross react with the antibody. Also, there will be both PGE2 and PGE3 in this mix and the latter reacts with the antibody. In short, this is NOT proof of true PGE2 only PG-like in terms of antibody binding.

Line 270. The use of one COX inhibitor, ibuprofen, in some experiments is not ideal. The authors should have used two or more to be certain that the effect seen relates to PG biosynthesis and not another result. Furthermore, the concentration of inhibitor used is vitally important to its selectivity of action. Too high leads to other side effects of the inhibitor. How was the dose used in these experiments determined? Did the authors try a dose response to determine the optimum concentration? This is really important and cannot be overlooked.

Blots in figure 3 and 5. The blots are presented as very high contrast where the lanes are invisible. The authors should show these as they really are so that any small bands etc can be seen alongside the bands of importance. The authors need to take care with potential excessive manipulation of images and show these as raw.

Have the authors checked for PGE2 synthesis by MS in crustacean cells not just transfected cells? Have they looked at the PG profile? (i.e. not just PGE2)

Overall, interesting and valuable work but with clear defects that require further work. If these additional experiments are carried out, then this will be a good paper for publication in PloS ONE.

6. PLOS authors have the option to publish the peer review history of their article (what does this mean?). If published, this will include your full peer review and any attached files.

Reviewer #1: No

Reviewer #2: No

---

## [Author Response · Author response to Decision Letter 0]

20 Mar 2021

Academic editor comments to author

The manuscript has been reviewed by two experts in the field; please find their comments appended (end of this email; attachment). Based on the reviewers’ comments as well as my own reading of the manuscript, I have decided for ‘Major Revision’ as there are two particular aspects where the data provided do not fully support the conclusions of this present manuscript.

Response: Thank you for the decision of major revision for this manuscript. We have provided a point-by-point response to all the comments from the editor as well as the reviewers as highlighted in blue and the revised manuscript was also modified with track changes.

A revised version should address all issues pointed out by the reviewers and in particular:

(1) include an independent experimental result to confirm the presence of PGE2 in this shrimp (e.g. by MS analysis);

Response: The authors verified the presence of prostaglandins in both P. monodon and P. vannamei using UPLC-HRMS/MS. S1 Data contains extract ion current and mass spectra of PGE2, PGD2 and PGF2alpha in intestines of wild P. monodon. S6 Data contains extract ion current and mass spectra of PGE2 and PGF2alpha in P. vannamei post-larvae. The content of the manuscript was also modified to include these data in line 287-290 and line 581-583. 

(2) include sufficient evidence or supporting data showing that PG anabolism can occur in this shrimp (e.g. by identification of relevant enzymes based on the shrimp genome).

Response: The presence of genes in the prostaglandin biosynthesis pathway in P. monodon has previously been established by our research team through the identification of full-length cytosolic phospholipase A2, COX, gluthathione-dependent prostaglandin D synthase, hematopoietic prostaglandin E synthase, prostaglandin E synthase-1,-2 and -3, prostaglandin F synthase and thromboxane A2 synthase from P. monodon ovary cDNA (Wimuttisuk et al., 2013). Additionally, sequence of gene downstream of this pathway, namely prostaglandin reductase 1, has also been identified in P. monodon by a separate research group (Prasertlux et al., 2011). Most of these gene sequences have been mapped onto the recently published P. monodon genome sequence (Uengwetwanich et al. 2021), which have been summarized in S2 Table. 

 Similar to P. monodon, genes in the prostaglandin biosynthesis pathway have also been identified P. vannamei. Because the effects of COX inhibition in the in vivo experiment was performed in P. vannamei, the information regarding genes in the prostaglandin biosynthesis pathway in P. vannamei was also included in S2 Table. 

We have also included the original uncropped and unadjusted images underlying all blots in Supporting Information (S1_Raw_Images.pdf).

Reviewers' comments:

Reviewer's Responses to Questions

Comments to the Author

Reviewer #1: 

Tobwar et al. describe the expression and characterization of cyclooxygenase (COX) in penaeid shrimp and identify two asparagine residues that must be glycosylated in the catalytically active enzyme. The authors provide evidence for the enzymatic function both in vitro and in vivo and highlight similarities between various species.

Overall, the work appears to be thorough and well considered. The results are of a high standard and support the conclusions. The paper is well-written and the overall conclusions are placed in the broader context of the potential benefit to the aquaculture industry should this system be leveraged in penaeid shrimp breeding.

Comments for the authors, minor issues to be addressed and typographical errors for correction are detailed in the attached document. This is an elegant piece of work that has been well reported and the authors are to be commended on the clarity of the writing and figures. I recommend publication following minor corrections.

Response: We thank Reviewer #1 and we have modified the manuscript based on the reviewers’ suggestion. We have responded to comments from Reviewer#1 in the table below.

Line Comment 

98 The wording “…analysis performed to obtain the amino acid sequence of PmCOX…” suggests that the sequence was previously unknown. As the sequence is in GenBank, I suggest rewording this to clearly state that sequencing was performed to confirm the correct sequence in the plasmid. 

Response: We have modified the wording based on the reviewer’s recommendation to “Sequencing was performed (1st-BASE, Malaysia) to confirm the correct sequence in the plasmid” (line 99-100).

122 Your use of the correct term ‘transformation into’ (instead of transformation with’) is a pleasure to see (and increasingly rare in our field!) – thank you! 

Response: Thank you so much for your comment.

155 CV for PGE2 EIA excellent – robust. 

Response: Thank you. 

160 Comment – pH 8.03 is highly specific; is there a reason why the .03 is noted (is the process that sensitive?) 

Response: Thank you for pointing this out. This section of the materials and method was written when we were troubleshooting the extraction protocol. We may have been overzealous in writing out details regarding the experimental conditions. We no longer measure the pH of BHT solution. Therefore, the wording has been removed from the revised manuscript. 

S1 Table/S1 Data I am glad to see the standard curves included here and note that the data/figure for PGE2 is included in S1 Data. For consistency, the standard curve for PGF2 should also be included in this file. 

Response: In the original draft of the manuscript, we did not perform EIA for PGF2alpha. It was detected mainly by UPLC-HRMS/MS analysis. Nevertheless, we have added a second tab in the supplementary data (now S2 Data) to show the standard curve for PGF2� as requested. The new data also showed that levels of PGF2� in cell culture media also increased in 293T cells expressing PmCOX compared to untransfected cells. 

271/272 Are you able to explain why haemolymph was sampled 48 h post-injection (for potential non-shrimp experts) instead of another timepoint? 

Response: We have also checked the levels of PGE2 in shrimp haemolymph at 6 and 24 h psi in a separate experiment using the same dosage of ibuprofen (see figure below). The reduction of PGE2 levels was negligible at 6 h and became more significant at 24 h psi. As our initial experiment with data collection at 48 h psi resulted in the highest percent inhibition, we decided to use that time point for our manuscript. 

294-297 Do the authors have any theories as to why the crustacean COX enzymes lack the C-terminal ER signals? 

Response: The study by Yuan and Smith (2015) proposed that the function of C-terminal ER signal was to increase the duration in which COX enzyme remains in the ER to increase the glycosylation efficiency. For mammalian COX2, this is crucial for the slow glycosylation process of residue N549. As the N549 residue was also absent in crustacean COXs, the presence of C-terminal ER signal was not required as other N-glycosylation sites can be glycosylated efficiently without the ER signal. 

303 “Predicted glycosylation sites, heme-binding residues, substrate-binding residues and an aspirin acetylation site were identified using multiple sequence alignment”. This wording suggests that these sites were predicted using the MSA, but the methods specify the use of NetNGlyc; I suggest rewording to avoid reader confusion. Also, does this mean that the additional residues of interest (heme- and substrate-binding residues and the acetylation site) were identified based on homology with other sequences in the alignment, since no other computational predication tool has been specified? 

Response: The reviewer’s understanding was correct. N-glycosylation sites were the only sequence predicted using computational prediction tool. Other sites, including heme- and substrate binding residues and acetylation sites were all identified based on homology. 

We have changed the wording of that sentence to “Predicted heme-binding residues, substrate-binding residues and an aspirin acetylation site were identified using multiple sequence alignment. N-glycosylation sites were predicted using NetNGlyc 1.0.” (Line 317-319). 

605 The distal and proximal histidines in PmCOX are identified here but no equivalent residues are mentioned for the other species under discussion here; for consistency with the rest of the paragraph, the equivalent residues should be mentioned along with an explanation of their significance in catalysis (as has been done for R120, Y335 and Y413). 

Response: We added information regarding the function of distal and proximal histidines, as well as its equivalent residues as follow. “Distal and proximal histidines, which are responsible for hydroperoxide reductase activities and coordinating the heme iron inside COX active site, are equivalent to residues H207 and H338 in OvCOX1 and residues H233 and H414 in PmCOX, respectively (16,51,52)” (Line: 621-624).

653-4 Comment: It may also be worth investigating whether glycosylation affects dimerization (as in COX2) and whether the role of glycosylation at N79 can be elucidated. 

Response: Thank you for your kind suggestion. We believe that N79 glycosylation is not involved in PmCOX protein turn over as the intensity of PmCOX-N79Q protein band was comparable to those of wild-type PmCOX (Fig. 5C). We also suspected that N79 glycosylation was not involved in COX dimerization as the enzymatic activity detected in PmCOX-N79 was comparable to those of wild-type COX (Figure 6B). If the lack of glycosylation on N79 residue reduced or disrupt COX dimerization, we expect that the enzymatic activity of PmCOX-N79Q mutant would be less than those of wild-type COX. Nevertheless, we will keep this in mind and determine the dimerization efficiency of COX-N79Q mutant using immunoprecipitation assay in future experiments. 

678 This line reads “Our preliminary data revealed that ARA is the preferred substrate for PmCOX, which is supported by the fact that other series of prostaglandins produced from EPA or other precursors have yet to be detected in crustaceans”. While the lack of evidence for other prostaglandins from EPA supports ARA as the preferred substrate (as does the specificity of fish COX1), the work reported here has not actually tested other substrates besides ARA and so, this sentence is misleading. Yes, ARA is a substrate for PmCOX (and yes, it is most likely preferred), but without the testing of other substrates, it cannot be conclusively stated based on this work said that ARA is preferred. Consider rewording this sentence slightly to avoid overstating this. 

Response: We have modified the wording to “As this study confirms that ARA serves as a substrate for PmCOX, the effects of dietary ARA on shrimp prostaglandin biosynthesis should be taken into consideration during the formulation of shrimp feed” in Line 696-698.

109/124 Inconsistent capitalization of CLUSTAL OMEGA; please make consistent.

Response: We have changed the text to “Clustal Omega” in both locations. 

176 Please switch hyphen to an en dash in (-80 degC)

Response: We have switched from hyphen to en-dash as suggested.

136,

143,

210, …

 Please capitalize ‘western’ in Western blot throughout.

Response: We have capitalized all Western blot throughout the manuscript.

190 Insert space between number and unit (2µM), and capitalize unit.

Response: We have added the space for 2 µm (micrometers), indicating the diameter of the beads in the Acclaim column. We cannot change it to 2 µM as suggested as it would be confused with 2 micromolar. (Line 191)

209 Change hyphen to en dash in -20degC

Response: We have switch from hyphen to en-dash as suggested.

216 Add space between number and unit (2mM)

Response: We have added the space for 2 mM EDTA as suggested (Line 217).

617 Typo: Please correct ‘Pichai’ to ‘Pichia’.

Response: We have corrected the typo as suggested (Line 635).

Reviewer #2: Major points

The major problem with this manuscript is the reliance of EIA in several crucial experiments to prove that PGE2 is actually present in this crustacean. I can see no evidence that this was checked/verified with MS or an equivalent specific method. 

Response: Our research team had previously confirmed the presence of PGE2 and PGF2� in ovaries of P. monodon via HPLC-MS (Wimuttisuk et al., 2013). However, we now realized that the lack of MS data PGE2 and PGF2� in penaeid shrimp in this manuscript detracts from the credibility of our findings. We have provided additional data for the UPLC-HRMS/MS analysis of prostaglandins in P. monodon (S1 Data) and P. vannamei (S6 Data) in the revised manuscript.

The authors have proven that there is active COX present but generation of PGs requires other enzymes that convert the intermediates generated by COX to PGs. Without such evidence this work CANNOT be published and the authors must carry out more experiments. 

Response: The presence of genes in the prostaglandin biosynthesis pathway in P. monodon has previously been established by our research team through the identification of full-length cytosolic phospholipase A2 (NCBI accession no. JN003878), COX (NCBI accession no. KF501342), gluthathione-dependent prostaglandin D synthase (NCBI accession no. JN003880), hematopoietic prostaglandin E synthase (NCBI accession no. JN003879), prostaglandin E synthase-1,-2 and -3 (NCBI accession no. JN003882 JN003883 and JN003881), prostaglandin F synthase (NCBI accession no. JN003884) and thromboxane A2 synthase (NCBI accession no. JN003885) from P. monodon ovary cDNA (Wimuttisuk et al., 2013). Additionally, a full-length sequence of prostaglandin reductase 1, which is downstream of the prostaglandin biosynthesis pathway, has also been identified in P. monodon by a separate research group (Prasertlux et al., 2011). Most of these gene sequences have recently been mapped onto the recently published P. monodon genome sequence (Uengwetwanich et al. 2021.) 

Similarly, the annotation of P. vannamei genome sequence also revealed the presence of genes in the prostaglandin biosynthesis pathway. We have summarized the information regarding prostaglandin biosynthesis genes in P. monodon and P. vannamei as shown in the S2 Table. 

Finally, the lack of dose response using ibuprofen is worrying and points to side effects of this inhibitor when at too high concentration. Perhaps some lower doses of inhibitor should be tried.

Response: We thank Reviewer#2 for your kind suggestion. Although it was not possible to conduct another ibuprofen trial at this time, we would like to present some of our already obtained data. This experiment involved the injection of 0.48 µg of ibuprofen in shrimp with 12 g body weight, which is the low concentration of ibuprofen used in Fig 7B. These shrimp were also infected with white spot syndrome virus. The experiment was performed in 3 separate tanks with four shrimp in each tank. For this experiment, shrimp haemolymph were harvested at 6 and 24 h post-injection instead of the 48 h post-injection used in the manuscript. Percent inhibitions were 2.1% and 21.9% at 6 and 24 h post-injection, respectively. Therefore, we believe that the reduction of PGE2 levels in haemolymph was due to the inhibition of PvCOX enzymatic activities and not the side effects of using too high concentration of ibuprofen. However, we refrained from adding this data into the manuscript as it detracts from the major point of this study. 

Minor points

Lines 23-4. This is not accurate description of the current status of COX identity in crustaceans (see lines 291 onwards for examples of COX already described in various crustaceans)

Response: We modified the sentence to “Although COX glycosylation requirement is well-characterized in many species, whether crustacean COXs require N-glycosylation for their enzymatic function have not been investigated.” (Line 22-24)

Line 29 The role of tunicamycin needs clarifying for the reader

Response: We expanded the sentence to clarify the function of both tunicamycin and endoglycosidase H as follow. “Incubation of PmCOX with endoglycosidase H treatment, which cleaves oligosaccharides from N-linked glycoproteins, reduced the molecular mass of PmCOX. Similarly, incubation of PmCOX-expressing cells with tunicamycin, which inhibits N-linked glycosylation, also produced PmCOX protein with lower molecular mass that those obtained from untreated cells, suggesting that PmCOX was N-glycosylated”.

Lines 69-70 This is irrelevant, so remove

Response: We removed the sentence as suggested.

Lines 150-155: This is a problem as EIA is not 100% specific in terms of antibody binding. Other PGs will cross react with the antibody. Also, there will be both PGE2 and PGE3 in this mix and the latter reacts with the antibody. In short, this is NOT proof of true PGE2 only PG-like in terms of antibody binding.

Response: The authors agree with the reviewer that EIA is not 100% specific in terms of antibody binding. However, the detection and quantification of PGE2 produced by 293T cells in this study were also supported by the data from UPLC-HRMS/MS analysis, which can distinguish between PGE2 and PGE3. For the quantification of PGE2 in shrimp haemolymph, which was based only on EIA, it was possible that a mixture of both PGE2 and PGE3 were present in shrimp haemolymph. However, while PGE2 have been detected in shrimp using HPLC-MS and UPLC-HRMS/MS both in this study and in previously published work, the presence of PGE3 has yet to be reported in any crustaceans. We believe that it is unlikely for PGE3 to be expressed at similar or higher levels than PGE2 in shrimp haemolymph. Nevertheless, we will keep this in mind and be more careful in our future analysis. We will also try to detect PGE3 in our shrimp samples using LC-HRMS/MS in future studies. 

Line 270. The use of one COX inhibitor, ibuprofen, in some experiments is not ideal. The authors should have used two or more to be certain that the effect seen relates to PG biosynthesis and not another result. Furthermore, the concentration of inhibitor used is vitally important to its selectivity of action. Too high leads to other side effects of the inhibitor. How was the dose used in these experiments determined? Did the authors try a dose response to determine the optimum concentration? This is really important and cannot be overlooked.

Response: Although we agreed with the reviewer that testing the dose response curve and choosing more than one COX inhibitor would be ideal to determine optimum inhibitor concentrations, we were limited by the restriction in the number of shrimp samples due to the requirement from the IACUC committee at our research institute. Therefore, we varied the inhibitor concentrations and the types of inhibitor in the in vitro experiment (which required less number of shrimp to be sacrificed) before selecting just one inhibitor for the in vivo experiment.

 In our initial experiment, we tested the ability of four COX inhibitors, including aspirin, indomethacin, ibuprofen and valeryl salicylate, to inhibit PGE2 production in our in vitro experiments using shrimp ovary homogenates. This resulted in 32, 52, 90 and 32% inhibition for aspirin, indomethacin, ibuprofen and valeryl salicylate, respectively. Ibuprofen was then selected for the dose-response curve in the in vitro assay as it resulted in the highest inhibition for PGE2 production. We have attached the data from the SoftMax Pro5 Program (Molecular Device, USA), which was used to extrapolate the value based on percent inhibition of ibuprofen in shrimp ovary homogenates (tissue concentration was 0.2 g/mL). The IC50 of ibuprofen was 184.5 µM in this organ. 

 For the data presented in this manuscript, we repeated the COX inhibitor treatment in shrimp haemolymph, which also led to similar conclusion that aspirin and ibuprofen treatment can inhibit PGE2 production (Fig. 7A). Therefore, the dosages obtained in the in vitro experiment was used as a base for the calculation of ibuprofen concentration in the in vivo experiment. Ibuprofen inhibition at 20 and 200 ng/mL in a mixture of haemolymph: sodium citrate at a 1:1 ratio (v/v) in the in vitro experiment were equivalent to 0.48 and 4.8 µg of ibuprofen in the 12-g shrimp (taken into account the whole blood volume that we could obtained from this shrimp). Moreover, a study by Alfaro-Montoya et al. (2015) injected ibuprofen at 0.1 µg/g shrimp to P. vannamei broodstock to induce ovarian development, which was higher than the low dosage of ibuprofen used in this study. As the effect of COX inhibitors was shown in both shrimp ovary homogenates, shrimp haemolymph and in the in vivo experiment, we believe that the observed reduction of PGE2 levels are not side effect from high dosages of COX inhibitors. 

Blots in figure 3 and 5. The blots are presented as very high contrast where the lanes are invisible. The authors should show these as they really are so that any small bands etc can be seen alongside the bands of importance. The authors need to take care with potential excessive manipulation of images and show these as raw.

Response: We apologize for the low background on the Western blot data. These data were obtained using the ChemiDoc XRS+ System (BioRad), which provided auto-contrast for the image to prevent oversaturation of the signal for quantification purpose. The image used in the original manuscript was obtained and placed in the manuscript without adjustment of color, brightness or contrast. However, based on the reviewer’s comments, we have now adjusted the image so that any small protein bands will become visible. The full blot image are included in supplementary data S1 Raw Images.

Have the authors checked for PGE2 synthesis by MS in crustacean cells not just transfected cells? Have they looked at the PG profile? (i.e. not just PGE2).

Response: Yes. We had previously detected PGE2 and PGF2� in ovaries of P. monodon via HPLC-MS (Wimuttisuk et al., 2013). Our current work also focused on the identification of prostaglandins in intestines of male P. monodon using UPLC-HRMS/MS. The analysis revealed the presence of PGE2, PGF2� and PGD2 in shrimp intestines as supported by the extracted ion chromatograms and mass spectra of PGE2, PGF2� and PGD2 (S1 Data). Additionally, PGE2 and PGF2� have also been identified in P. vannamei post-larvae (S6 Data). 

Overall, interesting and valuable work but with clear defects that require further work. If these additional experiments are carried out, then this will be a good paper for publication in PloS ONE.

Response: We thank your valuable suggestions that help improving the quality of our manuscript.

---

## [Decision Letter · Decision Letter 1]

5 Apr 2021

Biochemical characterization of the cyclooxygenase enzyme in penaeid shrimp

PONE-D-20-40833R1

Dear Dr. Wimuttisuk,

We’re pleased to inform you that your manuscript has been judged scientifically suitable for publication and will be formally accepted for publication once it meets all outstanding technical requirements.

Kind regards,

Andreas Hofmann

Academic Editor

PLOS ONE

Additional Editor Comments (optional):

Reviewers' comments:

Reviewer's Responses to Questions

**Comments to the Author**

1. If the authors have adequately addressed your comments raised in a previous round of review and you feel that this manuscript is now acceptable for publication, you may indicate that here to bypass the “Comments to the Author” section, enter your conflict of interest statement in the “Confidential to Editor” section, and submit your "Accept" recommendation.

Reviewer #1: All comments have been addressed

Reviewer #2: All comments have been addressed

2. Is the manuscript technically sound, and do the data support the conclusions?

Reviewer #1: Yes

Reviewer #2: Yes

3. Has the statistical analysis been performed appropriately and rigorously? 

Reviewer #1: N/A

Reviewer #2: N/A

4. Have the authors made all data underlying the findings in their manuscript fully available?

Reviewer #1: Yes

Reviewer #2: Yes

5. Is the manuscript presented in an intelligible fashion and written in standard English?

Reviewer #1: Yes

Reviewer #2: Yes

6. Review Comments to the Author

Reviewer #1: (No Response)

Reviewer #2: Thank you for answering the questions raised. I would suggest that you move some of the supplementary data into the main paper where they are crucial experiments.

7. PLOS authors have the option to publish the peer review history of their article (what does this mean?). If published, this will include your full peer review and any attached files.

Reviewer #1: No

Reviewer #2: No

---

## [Editor Report · Acceptance letter]

12 Apr 2021

PONE-D-20-40833R1 

Biochemical characterization of the cyclooxygenase enzyme in penaeid shrimp 

Dear Dr. Wimuttisuk:

I'm pleased to inform you that your manuscript has been deemed suitable for publication in PLOS ONE. Congratulations! Your manuscript is now with our production department. 

Kind regards, 

on behalf of

Associate Professor Andreas Hofmann 

Academic Editor

PLOS ONE